# The Sparse-Plus-Low-Rank Quasi-Newton Method for Entropic-Regularized Optimal Transport

**Chenrui Wang** [1]   **Yixuan Qiu** [1]

## Abstract

The entropic-regularized optimal transport (OT) has gained massive attention in machine learning due to its ability to provide scalable solutions for OT-based tasks. However, most of the existing algorithms, including the Sinkhorn algorithm and its extensions, suffer from relatively slow convergence in many cases. More recently, some second-order methods have been proposed based on the idea of Hessian sparsification. Despite their promising results, they have two major issues: first, there is limited theoretical understanding on the effect of sparsification; second, in cases where the transport plan is dense, Hessian sparsification does not perform well. In this paper, we propose a new quasi-Newton method to address these problems. First, we develop new theoretical analyses to understand the benefits of Hessian sparsification, which lays the foundation for highly flexible sparsification schemes. Then we introduce an additional low-rank term in the approximate Hessian to better handle the dense case. Finally, the convergence properties of the proposed algorithm are rigorously analyzed, and various numerical experiments are conducted to demonstrate its improved performance in solving large-scale OT problems.

## 1. Introduction

Optimal transport (OT), as originally described by Gaspard Monge, addresses the problem of moving a distribution (*e.g.*, a pile of sand) to match a target configuration (*e.g.*, a prescribed shape) while minimizing a cost, such as the total distance or effort required. In recent years, OT has received significant attention, due to its strong connections

with statistical modeling and machine learning tasks. In particular, OT provides a principled way to measure the similarity between probability distributions by considering the cost of transporting the mass between them (Villani et al., 2009), making it highly relevant in applications involving structured data or distributions with geometric properties. See Torres et al. (2021); Montesuma et al. (2024) for an overview of its applications in modern machine learning.

The discrete OT problem can be characterized by the following linear programming problem:

$$\min_{P \in \Pi(a,b)} \langle P, M \rangle,$$

$$\Pi(a,b) = \{P \in \mathbb{R}^{n \times m} : P\mathbf{1}_m = a, P^T\mathbf{1}_n = b, P \geq 0\},$$

where $a \in \mathbb{R}^n$, $b \in \mathbb{R}^m$ are two vectors satisfying $a^T\mathbf{1}_n = b^T\mathbf{1}_m = 1$, $a > 0$, $b > 0$, $M$ is a given cost matrix, and all inequality signs are elementwise.

More recently, the application of OT has experienced a significant boost due to the development of approximate solvers such as the entropic-regularized OT. Popularized by Cuturi (2013), the entropic-regularized OT incorporates an entropic regularization $h(P) = \sum_{i=1}^{n} \sum_{j=1}^{m} P_{ij}(1 - \log P_{ij})$ into the OT problem:

$$\min_{P \in \Pi(a,b)} \langle P, M \rangle - \eta h(P). \tag{1}$$

This regularization significantly reduces the computational cost of OT based on the Sinkhorn–Knopp algorithm (Yule, 1912; Sinkhorn, 1964), thus unlocking its potential in large-scale problems. Along this direction, many extensions of the Sinkhorn algorithm have also been proposed (Altschuler et al., 2017; Dvurechensky et al., 2018; Guminov et al., 2021; Lin et al., 2022). As a result, OT is increasingly applied to solve a wide range of challenges in fields such as image processing, graphics, and machine learning (Peyré et al., 2019).

However, the computation of problem (1) is still a major challenge. For example, the Sinkhorn algorithm generally requires a large number of iterations to converge, especially when the regularization parameter is small. Other first-order methods that extend the Sinkhorn algorithm also show relatively slow convergence. To this end, another approach to

---

[1]School of Statistics and Data Science, Shanghai University of Finance and Economics, Shanghai, China. Correspondence to: Yixuan Qiu <qiuyixuan@sufe.edu.cn>.

*Proceedings of the 42$^{nd}$ International Conference on Machine Learning*, Vancouver, Canada. PMLR 267, 2025. Copyright 2025 by the author(s).

solving entropic-regularized OT is to study the dual problem of (1), *i.e.*, maximizing a function $\mathcal{L}(\alpha, \beta)$, $\alpha \in \mathbb{R}^n$, $\beta \in \mathbb{R}^m$, where

$$\mathcal{L}(\alpha, \beta) = \alpha^T a + \beta^T b$$
$$- \eta \sum_{i=1}^{n} \sum_{j=1}^{m} \exp\{\eta^{-1}(\alpha_i + \beta_j - M_{ij})\}. \quad (2)$$

Once an optimal solution $(\alpha^*, \beta^*)$ to problem (2) is sought, the primal optimal solution to (1), denoted by $T^* \in \mathbb{R}^{n \times m}$, can be obtained as $T_{ij}^* = \exp\{\eta^{-1}(\alpha_i^* + \beta_j^* - M_{ij})\}$. Since $\mathcal{L}(\alpha, \beta) = \mathcal{L}(\alpha + c\mathbf{1}_n, \beta - c\mathbf{1}_m)$ for all $c \in \mathbb{R}$, we can remove the redundant degree of freedom by setting $\beta_m = 0$ globally. Then eventually, solving entropic-regularized OT is equivalent to solving

$$\min_{x \in \mathbb{R}^{n+m-1}} f(x), \quad (3)$$

where

$$f(x) = -\mathcal{L}(\alpha, \beta) = \eta \sum_{i=1}^{n} \sum_{j=1}^{m} \exp\{\eta^{-1}(\alpha_i + \beta_j - M_{ij})\}$$
$$- \alpha^T a - \beta^T b,$$

and $x = (\alpha_1, \ldots, \alpha_n, \beta_1, \ldots, \beta_{m-1})^T$. Clearly, (3) is a smooth and unconstrained convex optimization problem, which brings the possibility to use second-order methods for acceleration.

However, the classical Newton method is not a realistic approach, since the Hessian matrix of $f(x)$ is a dense matrix of the size $(n + m - 1) \times (n + m - 1)$, leading to an $O((n+m)^3)$ computational cost in computing the Newton direction. More recently, Tang et al. (2024) and Tang & Qiu (2024) tackle this problem based on the idea of Hessian sparsification, *i.e.*, approximating the true Hessian matrix by sparse matrices, thus leading to efficient sparse linear systems to compute the search directions. This line of works show some promising results, but two major issues remain to be solved: first, there is limited theoretical understanding on the effect of sparsification; second, in cases where the true Hessian matrix is relatively dense, Hessian sparsification does not perform well.

In this paper, we mainly target on resolving the two issues above, and propose a new quasi-Newton method to solve entropic-regularized OT. First, we provide new theoretical analyses on the sparsified Hessian matrices, and show that sparsification brings various benefits. The results also motivate a class of flexible sparsification schemes that substantially generalize existing methods. Second, we propose a new model to better approximate the true Hessian matrix, which combines Hessian sparsification with low-rank approximation. We demonstrate that this method greatly enhances the algorithm's performance in scenarios where the

Hessian matrix is relatively dense. Rigorous convergence analysis is provided to support the application of the proposed algorithm in large-scale OT problems. An efficient implementation of the method is included in the RegOT Python package[1].

**Contribution** Our main contribution compared to prior art is summarized as follows:

1. New theoretical results are developed to understand the mechanism of Hessian sparsification. Such a theoretical understanding also guarantees that a broad range of sparsification schemes enjoy desirable properties.

2. A new quasi-Newton method is proposed to solve entropic-regularized OT that combines the advantages of Hessian sparsification and low-rank approximation, achieving fast convergence speed with low computational cost.

3. We provide convergence guarantees for the proposed method, and conduct extensive numerical experiments to demonstrate its performance.

**Notation** Throughout this article we adopt the following notation. For $n \in \mathbb{N}$, denote $[n] := \{1, 2, \ldots, n\}$. Given a matrix $A \in \mathbb{R}^{n \times m}$ and a vector $v \in \mathbb{R}^m$, we use $\tilde{A} \in \mathbb{R}^{n \times (m-1)}$ to represent the first $(m-1)$ columns of $A$, and $\tilde{v} \in \mathbb{R}^{m-1}$ to represent the first $(m-1)$ elements of $v$; inequality signs such as $A > 0$ and $v < 0$ are all elementwise. We use $\lambda_{\max}(\cdot)$ and $\lambda_{\min}(\cdot)$ to represent the largest and smallest eigenvalues of real symmetric matrices, respectively. For a matrix $A$, let $A_{i\cdot}$ be the vector of the $i$-th row of $A$, and $A_{\cdot j}$ be the vector of the $j$-th column of $A$.

## 2. Related Work

**OT in machine learning** Optimal transport has emerged as a powerful mathematical framework with diverse applications in machine learning (Torres et al., 2021; Montesuma et al., 2024). It is widely used for tasks such as domain adaptation (Courty et al., 2017), generative modeling (Arjovsky et al., 2017; Genevay et al., 2018; Huynh et al., 2021), clustering (Laclau et al., 2017), and cross-domain alignment (Chen et al., 2020), among many others.

**Solving entropic-regularized OT** There is a rich collection of algorithms developed to solve entropic-regularized OT. Following the seminal work Cuturi (2013) that popularizes the Sinkhorn–Knopp algorithm (Yule, 1912; Sinkhorn, 1964), many extension methods are proposed (Altschuler et al., 2017; Dvurechensky et al., 2018; Guminov et al., 2021; Lin et al., 2022). In addition to first-order methods,

---

[1] https://github.com/yixuan/regot-python

Brauer et al. (2017); Tang et al. (2024); Tang & Qiu (2024) consider Newton-type second-order methods to solve the dual problem of entropic-regularized OT. Another direction, as in the importance sparsification approach (Li et al., 2023), accelerates computation by constructing a sparse approximation of the kernel matrix, thereby significantly reducing the cost of Sinkhorn iterations.

**Quasi-Newton methods** Quasi-Newton methods are a class of optimization techniques designed to efficiently solve large-scale smooth optimization problems (Nocedal & Wright, 2006). Unlike the classical Newton methods, they approximate the Hessian matrix by simpler structures in computing the search directions, thus reducing computational complexity while retaining rapid convergence. These features make them highly effective for machine learning and optimization tasks involving high-dimensional data. For example, Cuturi & Peyré (2018) suggests using the limited-memory Broyden–Fletcher–Goldfarb–Shanno (L-BFGS) method (Liu & Nocedal, 1989) to solve the dual problem of entropic-regularized OT.

## 3. Understanding Hessian Sparsification

### 3.1. Background

Recall that our main objective is to solve the unconstrained convex optimization problem (3). It is known that $f(x)$ has some good properties:

1. $f(x)$ is strictly convex, so if (3) has a solution, then the solution is unique.

2. The gradient of $f(x)$, denoted by $g(x) := \nabla f(x)$, has a closed-form expression:

$$g(x) = \begin{bmatrix} T\mathbf{1}_m - a \\ \tilde{T}^T\mathbf{1}_n - \tilde{b} \end{bmatrix}, \ T = T(\alpha, \beta),$$

where the free variables are $x = (\alpha^T, \tilde{\beta}^T)^T$, $\beta = (\tilde{\beta}^T, 0)^T$, $T(\alpha, \beta)$ is an $n \times m$ matrix with elements $[T(\alpha, \beta)]_{ij} = \exp\{\eta^{-1}(\alpha_i + \beta_j - M_{ij})\}$, and recall that $\tilde{T}$ means removing the last column of $T$.

3. $f(x)$ is twice differentiable, and the Hessian matrix $H(x) := \nabla^2 f(x)$ also has a simple expression:

$$H(x) = \eta^{-1} \begin{bmatrix} \mathbf{diag}(T\mathbf{1}_m) & \tilde{T} \\ \tilde{T}^T & \mathbf{diag}(\tilde{T}^T\mathbf{1}_n) \end{bmatrix}. \quad (4)$$

The classical Newton method solves (3) by generating a sequence of iterates $\{x_k\}$ based on the update rule $x_{k+1} = x_k - \alpha_k H_k^{-1} g_k$, where $g_k = g(x_k)$, $H_k = H(x_k)$, and $\alpha_k > 0$ is the step size at iteration $k$. However, solving $H_k^{-1} g_k$ for a dense Hessian matrix $H_k$ has a computational

cost at the order of $O((n + m)^3)$, which is too demanding for large-scale OT problems. Therefore, several Hessian sparsification methods have been proposed to approximate $H_k$ by some sparse matrices, which lead to significantly reduced computational costs.

The SNS algorithm (Tang et al., 2024) sparsifies the Hessian matrix by a thresholding rule. Specifically, any entry in the Hessian matrix smaller than a constant $\rho$ is truncated to zero. However, although this procedure preserves the symmetry and diagonal dominance of the Hessian matrix, there is no guarantee on the positive definiteness of the sparsified Hessian. Specifically, 0 might be included in one of the Gershgorin discs, admitting $\lambda = 0$ as a possible eigenvalue.

The SSNS algorithm (Tang & Qiu, 2024) takes a similar but different approach, which only sparsifies the off-diagonal elements of the Hessian matrix. Meanwhile, it controls the row-wise and column-wise approximation errors. This method provides guarantees on the positive definiteness, but it is not flexible to design the sparsity pattern. In particular, the density after sparsification is unknown in advance, thus making it difficult to control the computational cost. To this end, in Section 3.2 we carefully analyze the eigenvalue structure of the sparsified Hessian, which provides various new insights on the consequence of Hessian sparsification.

### 3.2. Eigenvalue structure of the sparsified Hessian

As can be seen from (4), the true Hessian matrix at $x = (\alpha^T, \tilde{\beta}^T)^T$ is determined by the matrix $T = T(\alpha, \beta)$. Therefore, a natural way to sparsifying $H(x)$ is to sparsify the $T$ matrix. Following the method proposed in Tang & Qiu (2024), we consider the *off-diagonal* sparsification framework as given in Definition 3.1.

**Definition 3.1** (Sparsification scheme)**.** A sparsification scheme is defined by a set of coordinates $\Omega \subseteq \bar{\Omega} = \{(i, j) : i \in [n], j \in [m-1]\}$. In particular, the sparsified matrix $\tilde{T}_\Omega$ has elements

$$(\tilde{T}_\Omega)_{ij} = \begin{cases} \tilde{T}_{ij}, & (i,j) \in \Omega, \\ 0, & (i,j) \notin \Omega, \end{cases}$$

and the sparsified Hessian matrix is given by

$$H_\Omega = H_\Omega(x) = \eta^{-1} \begin{bmatrix} \mathbf{diag}(T\mathbf{1}_m) & \tilde{T}_\Omega \\ \tilde{T}_\Omega^T & \mathbf{diag}(\tilde{T}^T\mathbf{1}_n) \end{bmatrix}.$$

Clearly, $H_\Omega$ depends on the current variable $x$, but we will omit it for brevity if no confusion is caused. It is also worth noting that the diagonal elements of $H_\Omega$ remain unchanged compared to $H = H(x)$, which are computed from the original $T$ instead of the sparsified $\tilde{T}_\Omega$. This structure is crucial for our theoretical analysis.

To gain insights on the effect of sparsification, we first consider the process of *incremental sparsification*: removing

exactly two symmetric elements from the current approximate Hessian $H_{\Omega_0}$, resulting in another matrix $H_{\Omega_1}$, where $\Omega_0$ and $\Omega_1$ only differ by one element, and $\Omega_1 \subset \Omega_0$. Then the main theoretical result of this section, Theorem 3.3, claims that $H_{\Omega_1}$ strictly decreases the condition number of $H_{\Omega_0}$.

**Assumption 3.2.** For a sparsification scheme $\Omega \subseteq \bar{\Omega}$, there exists a positive integer $p > 0$ such that the $p$-th power of $H_\Omega$ has strictly positive entries, *i.e.*, $(H_\Omega)^p > 0$.

**Theorem 3.3.** *Given two sparsification schemes $\Omega_0, \Omega_1 \subseteq \bar{\Omega}$, suppose that $\Omega_1 \subset \Omega_0$ and they only differ by one element. If Assumption 3.2 holds for $\Omega_1$, then we have*

$$\lambda_{\max}(H_{\Omega_1}) < \lambda_{\max}(H_{\Omega_0}),$$
$$\lambda_{\min}(H_{\Omega_1}) > \lambda_{\min}(H_{\Omega_0}),$$

*which implies that $H_{\Omega_1}$ strictly decreases the condition number of $H_{\Omega_0}$.*

Suppose that we have chosen a specific sparsification scheme $\Omega \subseteq \bar{\Omega}$ that satisfies Assumption 3.2, and then there must exist a sequence of sparsification schemes $\{\Omega_t : t \in [T]\}$ that starts with $\Omega_1 = \bar{\Omega}$ and ends with $\Omega_T = \Omega$, satisfying $\Omega_{t+1} \subset \Omega_t, \forall t \in [T-1]$, and $\Omega_t$ and $\Omega_{t+1}$ only differ by one element. Since Assumption 3.2 holds for $\Omega = \Omega_T$, Lemma B.4 implies that it also holds for all sparsification schemes in the sequence, as $\Omega_T \subseteq \Omega_t, \forall t \in [T]$. Then by applying Theorem 3.3 repeatedly to $\Omega_t$ and $\Omega_{t+1}$ for $t = 1, \ldots, T-1$, we obtain the following corollary.

**Corollary 3.4.** *For any sparsification scheme $\Omega \subseteq \bar{\Omega}$ satisfying Assumption 3.2, the corresponding sparsified Hessian matrix $H_\Omega$ has the following properties:*

$$\lambda_{\max}(H_\Omega) \leq \lambda_{\max}(H),$$
$$\lambda_{\min}(H_\Omega) \geq \lambda_{\min}(H),$$

*where $H = H_{\bar{\Omega}} = H(x)$. The equalities hold if and only if $\Omega = \bar{\Omega}$.*

### 3.3. Numerical Verification

We numerically verify Theorem 3.3 and Corollary 3.4 through a simple experiment that illustrates how eigenvalues evolve during the sparsification process. Specifically, we generate a random matrix $T \in \mathbb{R}^{n \times m}$ with entries $T_{ij} \overset{iid}{\sim} \text{Unif}(0, 1)$, and construct the corresponding Hessian matrix $H$. We then iteratively set one nonzero element of $T$ to zero at each step, recalculating the minimum and maximum eigenvalues of $H$ at each iteration, until only the first row and first column of $T$ remain nonzero (see Theorem 4.1 for the rationale of this setting). We repeat this procedure five times, and illustrate the change of eigenvalues and condition number of the sparsified Hessian in Figure 1.

The results clearly demonstrate that during the incremental sparsification process, the minimum eigenvalue monotonically increases, while the maximum eigenvalue monotonically decreases, which is consistent with our theoretical predictions.

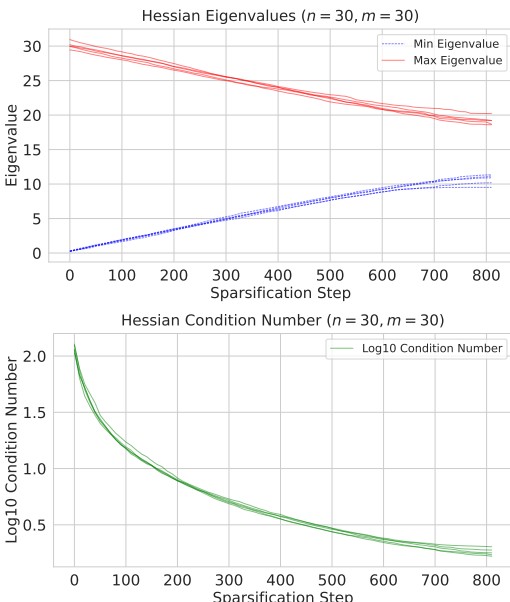

*Figure 1.* Plot of minimum and maximum eigenvalues of the sparsified Hessian matrix during the sparsification process.

### 3.4. Summary

We summarize this section by making a few remarks on the significance of Theorem 3.3 and Corollary 3.4:

1. Since the true Hessian matrix is positive definite, Corollary 3.4 implies that *any* valid sparsification scheme also maintains positive definiteness. This is crucial for computing the search directions, as the approximate Hessian also needs to be inverted.

2. The theorems indicate that the sparsified Hessian has a smaller condition number, which guarantees better numerical stability in solving linear systems. Even better, a smaller condition number makes iterative sparse linear solvers (e.g. conjugate gradient method) converge faster.

3. The controlled condition number is essential for the theoretical analysis of quasi-Newton methods; see for example Section 3.2 of Nocedal & Wright (2006).

4. Corollary 3.4 is valid for almost *any* sparsification scheme. This theoretical guarantee allows for highly flexible algorithmic designs, which greatly generalizes prior works on this direction.

5. The only requirement is Assumption 3.2, which is very weak: it can be satisfied with *extremely sparse* matrices. In Section 4.2, we provide a constructive method to let sparsification schemes easily satisfy this assumption.

# 4. The Sparse-Plus-Low-Rank Method

## 4.1. Overview

Despite the various benefits as explained in Section 3, the Hessian sparsification method has one intrinsic limitation: it highly relies on the sparsity property of the true Hessian matrix. When the true Hessian is dense, no sparsified version would perform well, and we need better models to approximate the Hessian matrix. To this end, we introduce the sparse-plus-low-rank (SPLR) approach that improves the approximation by adding low-rank terms to the sparsified Hessian. At a high level, we adopt the quasi-Newton framework to solve (3) based on the update rule

$$x_{k+1} = x_k - \alpha_k B_k^{-1} g_k,$$

where $B_k$ is an approximation to the true Hessian matrix $H_k$, and $\alpha_k > 0$ is a suitable step size that satisfies the Wolfe conditions given in (9). In our method, $B_k$ consists of three parts:

$$B_k = H_\Omega + (auu^T + bvv^T) + \tau I, \qquad (5)$$

where $H_\Omega$ is the sparsified Hessian matrix according to some sparsification scheme $\Omega$, $auu^T + bvv^T$ is a rank-two approximation term, and $\tau > 0$ is a shift parameter. Note that all these terms may vary with the iteration number $k$.

The intuition behind (5) is that when $H_k$ is truly close to a sparse matrix, $H_\Omega$ would be able to capture most of its information. And when this is not the case, the low-rank term can then compensate for the possibly dense difference $H_k - H_\Omega$. The shift term $\tau I$ is used to enhance the numerical stability when inverting the approximate Hessian matrix. Overall, $B_k$ is expected to perform as well as existing sparse Newton methods, and to show its advantage when $H_k$ is relatively dense.

In Algorithm 1, we first present our main SPLR algorithm to solve entropic-regularized OT, and then elaborate its details in subsequent sections, such as the choice of $\Omega$, the specification of $(a, b, u, v)$, etc.

## 4.2. Sparsification with density

One of the most important ingredients of the proposed SPLR algorithm is the choice of the sparsification scheme $\Omega$ in (5). First recall that an important condition for Corollary 3.4 to hold is that $\Omega$ satisfies Assumption 3.2. Below we first show that there is a "minimal" scheme $\Omega^*$ meeting this requirement, and then any scheme containing $\Omega^*$ would also satisfy Assumption 3.2.

---

**Algorithm 1** The Sparse-plus-low-rank quasi-Newton method for entropic-regularized OT

---

**Input:** Initial point $x_0$, maximum density $\rho_{\max} \in [0, 1]$, maximum shift $\tau_{\max} > 0$, stopping criterion $\varepsilon_{tol} > 0$

**Output:** $x_k$

1: Set $\rho_{\min} = 0.01 \cdot \rho_{\max}, \rho_0 = 0.1 \cdot \rho_{\max}, \Omega_0 = \Omega^*(\rho_0)$
2: Compute $f_0 = f(x_0), g_0 = g(x_0), H_0 = H(x_0)$
3: Sparsify $H_0$ according to scheme $\Omega_0$ to obtain $H_{\Omega_0}^0$
4: Set $\tau_0 = \min\{\tau_{\max}, \|g_0\|\}$
5: Compute $d_0 = -(H_{\Omega_0}^0 + \tau_0 I)^{-1} g_0$
6: Select the step size $\alpha_0$ and update $x_1 = x_0 + \alpha_0 d_0$
7: **for** $k = 1, 2, \ldots$ **do**
8:     Compute $f_k = f(x_k), g_k = g(x_k), H_k = H(x_k)$
9:     **if** $\|g_k\| < \varepsilon_{tol}$ **then**
10:        **return** $x_k$
11:     **end if**
12:     Update $\rho_k$ according to (8)
13:     Compute $\Omega_k = \Omega^*(\rho_k)$ with Algorithm 2, and sparsify $H_k$ to obtain $H_{\Omega_k}^k$
14:     Compute $s_{k-1} = x_k - x_{k-1}, y_{k-1} = g_k - g_{k-1}$
15:     Compute $a, b, u, v$ according to (7)
16:     Let $L = \begin{cases} auu^T + bvv^T, & \text{if } y^T s > 10^{-6}\|y\|^2 \\ O, & \text{otherwise} \end{cases}$
17:     Update $\tau_k = \min\{\tau_{\max}, \|g_k\|\}$
18:     Compute $d_k = -B_k^{-1} g_k$, where $B_k = H_{\Omega_k}^k + L + \tau_k I$
19:     Select the step size $\alpha_k$ and update $x_{k+1} = x_k + \alpha_k d_k$
20: **end for**

---

**Theorem 4.1.** *Define $\Omega^* = \{(i, j) : i = 1 \text{ or } j = 1, i \in [n], j \in [m-1]\}$, and let $H_{\Omega^*}$ be the corresponding sparsified Hessian matrix. Then $\Omega^*$ satisfies Assumption 3.2 with $(H_{\Omega^*})^4 > 0$.*

*Remark* 4.2. The choice "$i = 1$ or $j = 1$" in the definition of $\Omega^*$ is not essential. One can use "$i = i^*$ or $j = j^*$" for any fixed values of $i^*$ and $j^*$. $\Omega^*$ basically means keeping one row and one column of $\tilde{T}$ and setting other elements to zero.

Theorem 4.1 shows that it is reasonable to use $\Omega^*$ as our sparsification scheme. However, $\Omega^*$ may contain too few elements to provide a good approximation to $H$. For better performance, we keep a fixed proportion $\rho$ of the largest elements of $\tilde{T}$, leading to a sparsification scheme $\Omega(\rho)$, where $\rho \in [0, 1]$ represents the density. To satisfy Assumption 3.2, we take the union of $\Omega^*$ and $\Omega(\rho)$, denoted by $\Omega^*(\rho) := \Omega(\rho) \cup \Omega^*$. Then we can show that $\Omega^*(\rho)$ also satisfies Assumption 3.2 according to Lemma B.4.

Formally, first define an operator `select_large`$(\tilde{T}, \rho)$, which takes an $n \times (m-1)$ matrix $\tilde{T}$ and a density $\rho$ as inputs, and outputs a set $\Omega$ consisting of the coordinates of the largest $\lfloor \rho n(m-1) \rfloor$ elements in $\tilde{T}$. Then the sparsification scheme with a given density is obtained via Algorithm 2.

**Algorithm 2** Sparsification scheme with a given density

---

**Input:** Dual variable vector $x = (\alpha^T, \tilde{\beta}^T)^T$, density parameter $\rho \in [0, 1]$
**Output:** Sparsification scheme $\Omega^*(\rho)$
 1: Compute $T = T(\alpha, \beta)$
 2: Compute $\Omega(\rho) = \texttt{select\_large}(\tilde{T}, \rho)$
 3: Set $\Omega^*(\rho) = \Omega(\rho) \cup \Omega^*$

---

### 4.3. Low-rank terms

To enhance the approximation quality when the true Hessian is not well captured by sparsification alone, we incorporate a low-rank correction term. Specifically, suppose that we are at the $(k + 1)$-th iteration of the Newton-type optimization procedure. We approximate $H_{k+1}$ by a matrix $B_{k+1}$ of the form:

$$H_{k+1} \approx B_{k+1} := H_\Omega^{k+1} + auu^T + bvv^T + \tau_{k+1}I,$$

where $H_\Omega^{k+1}$ is the sparsified version of $H_{k+1}$ according to a scheme $\Omega$, and $a, b \in \mathbb{R}, u, v \in \mathbb{R}^{n+m-1}$ are to be specified later. Motivated by various quasi-Newton methods, especially the Broyden–Fletcher–Goldfarb–Shanno (BFGS) update rule, we determine $a, b, u, v$ by the secant equation, namely the first-order approximation of $g$ at $x_{k+1}$:

$$g(x) = g_{k+1} + H_{k+1}(x - x_{k+1}) + O(\|x - x_{k+1}\|^2).$$

Replace $x$ with $x_k$, and we get

$$g_k = g_{k+1} + H_{k+1}(x_k - x_{k+1}) + O(\|x_k - x_{k+1}\|^2).$$

Let $s_k = x_{k+1} - x_k$ and $y_k = g_{k+1} - g_k$. By ignoring the remainder term, we have $y_k \approx H_{k+1}s_k$. So it is reasonable to require the approximation of $H_{k+1}$, namely $B_{k+1}$, to satisfy:

$$y_k = B_{k+1}s_k = (H_\Omega^{k+1} + auu^T + bvv^T + \tau_{k+1}I)s_k. \quad (6)$$

One solution to (6) is:

$$u = y_k, \quad v = (H_\Omega^{k+1} + \tau_{k+1}I)s_k,$$
$$a = \frac{1}{y_k^T s_k}, \quad b = -\frac{1}{s_k^T(H_\Omega^{k+1} + \tau_{k+1}I)s_k}. \quad (7)$$

Although such a modification makes the approximation dense again, its inverse can be computed very conveniently with sparse or vector-based arithmetics (see for example Section 6.1 of Nocedal & Wright, 2006):

$$B_{k+1}^{-1} = U^T(H_\Omega^{k+1} + \tau_{k+1}I)^{-1}U + \xi_k s_k s_k^T,$$

where $\xi_k = 1/(y_k^T s_k)$ and $U = I - \xi_k y_k s_k^T$. Since $(H_\Omega^{k+1} + \tau_{k+1}I)$ is still a sparse and positive definite matrix, linear systems associated with it can be efficiently solved via either direct methods such as the sparse Cholesky decomposition, or iterative methods such as the conjugate gradient method.

### 4.4. Practical implementation

**Shift parameter** The shift parameter $\tau_k$ in Algorithm 1 is not necessary for theoretical analysis, and in fact, one can globally set $\tau_k \equiv 0$ without breaking the algorithm. However, in practical implementation, it brings various benefits, for example, stabilizing the linear system $B_k^{-1}g_k$, and potentially accelerating iterative linear solvers such as the conjugate gradient method. This is because $H_\Omega^k + \tau_k I$ has a smaller condition number than $H_\Omega^k$.

To avoid introducing a large approximation error, we dynamically set $\tau_k$ to be the current gradient norm $\|g_k\|$, so that the $\tau_k I$ term in (5) is negligible when $x_k$ is close to the optimum. An additional safeguard is to set a maximum shift $\tau_{\max}$, in case $\|g_k\|$ is too large at the beginning. So overall, we take $\tau_k = \min\{\tau_{\max}, \|g_k\|\}$ in each iteration.

**Adaptive density selection** Thanks to the theoretical guarantee presented in Corollary 3.4, we have a high degree of freedom to design the sparsification scheme in each iteration. In our implementation, the density parameter $\rho_k$ varies according to $\|g_k\|$. If $\|g_k\|$ decreases compared to the previous iteration, it means that the previous $B_{k-1}$ potentially provides a good approximation to $H_{k-1}$, so we can try a more sparse $H_\Omega^k$ in the current iteration, thus accelerating the search direction computation. Otherwise, we should increase the density to obtain a more precise approximation to $H_k$. Based on this idea, the update rule for $\rho_k$ is:

$$\rho_k = \begin{cases} \max\{\rho_{\min}, 0.99\rho_{k-1}\}, & \text{if } \|g_k\| < \|g_{k-1}\| \\ \min\{\rho_{\max}, 1.1\rho_{k-1}\}, & \text{otherwise} \end{cases}. \quad (8)$$

**Line search method** Finally, we use the Moré–Thuente line search algorithm (Moré & Thuente, 1994) to compute the step size $\alpha_k > 0$ that satisfies the Wolfe conditions:

$$f(x_k + \alpha_k d_k) \leq f(x_k) + c_1\alpha_k(g_k^T d_k),$$
$$[g(x_k + \alpha_k d_k)]^T d_k \geq c_2(g_k^T d_k), \quad (9)$$

where $0 < c_1 < 1/2$ and $c_1 < c_2 < 1$ are pre-specified constants.

## 5. Convergence Analysis

In this section, we prove that the proposed Algorithm 1 enjoys a global convergence property, and the convergence rate is at least linear. Much of the theory has been developed in classical quasi-Newton literature such as Byrd et al. (1987), but the key challenge here is to verify certain properties of the approximate Hessian matrix $B_{k+1}$.

In particular, the key to the convergence of quasi-Newton methods is the condition number of $B_{k+1}$ in each iteration. Therefore, it is crucial to bound both the smallest and

largest eigenvalues of $B_{k+1}$ for every $k$. We then have the following key findings.

**Theorem 5.1.** *Assume that there is a closed set $D$ such that*

$$\lambda_{\min}(H(x)) \geq L, \quad \lambda_{\max}(H(x)) \leq U$$

*for some constants $L, U > 0$ and all $x \in D$, and that $(1-t)x_k + tx_{k+1} \in D$ for all $t \in [0,1]$. Then*

$$\lambda_{\min}(B_{k+1}) \geq (2 + 3U/L)^{-1}L,$$
$$\lambda_{\max}(B_{k+1}) \leq 2U + \tau_{\max}.$$

After bounding the eigenvalues of $B_{k+1}$, we then obtain the global convergence of the proposed algorithm.

**Corollary 5.2.** *Let $x_0$ be an arbitrary initial value, and $\{x_k\}$ be generated by Algorithm 1. Then*

$$\lim_{k \to \infty} \|g(x_k)\| = 0.$$

Finally, we show that Algorithm 1 at least has a linear convergence rate.

**Theorem 5.3.** *Let $f^*$ be the optimal value of $f(x)$. Then for all $k \geq 1$, there is a constant $0 < r < 1$ such that*

$$f(x_{k+1}) - f^* \leq r[f(x_k) - f^*].$$

Although the theory only gives a linear convergence rate, our numerical experiments in Section 6 suggest that in many cases, the proposed SPLR algorithm achieves super-linear-like convergence speed.

## 6. Numerical Experiments

In this section, we evaluate the performance of the proposed SPLR algorithm via a series of numerical experiments, and compare SPLR with a number of widely-used algorithms for solving entropic-regularized OT: 1. the Sinkhorn algorithm (equivalent to block coordinate descent, BCD); 2. the adaptive primal-dual accelerated gradient descent (APDAGD, Dvurechensky et al., 2018); 3. L-BFGS; 4. the Newton method; 5. the SSNS algorithm (Tang & Qiu, 2024).

We consider both synthetic and realistic OT settings, and fix the regularization parameter to be $\eta = 0.001$. To make $\eta$ comparable for different problems, we normalize all cost matrices to have a unit infinity norm, *i.e.*, $M \leftarrow M/\|M\|_\infty$. We use the gradient norm $\|g(x_k)\|$ to measure the optimization error of the current iterate $x_k$. Additional test examples with different cost matrix settings and $\eta$ values are given in Appendix A.2 and Appendix A.3, respectively. The experiments in this section can be reproduced using the code on our Github repository[2].

[2] https://github.com/Aoblex/numerical-experiments

### 6.1. Synthetic data

We first consider two synthetic datasets that have been analyzed by existing sparse Newton methods (Tang et al., 2024; Tang & Qiu, 2024):

**Synthetic I:** $M = (M_{ij})$ has uniformly distributed entries, *i.e.*, $M_{ij} \overset{iid}{\sim} \text{Unif}(0,1)$, and $a = n^{-1}\mathbf{1}_n, b = m^{-1}\mathbf{1}_m$.

**Synthetic II:** This setting approximates OT between two continuous distributions: an exponential distribution with mean one, and a normal mixture distribution $0.2 \cdot N(1, 0.2) + 0.8 \cdot N(3, 0.5)$. The vectors $a$ and $b$ contain the discretized density function values of the two distributions, computed in the following way: let $x_i = 5(i-1)/(n-1), i \in [n]$, and $y_j = 5(j-1)/(m-1), j \in [m]$ be equally spaced points on $[0, 5]$, and let $f_1$ and $f_2$ be the density functions of the two distributions, respectively. Then set $\bar{a}_i = f_1(x_i), \bar{b}_j = f_2(y_j)$, $a_i = \bar{a}_i/(\sum_{k=1}^n \bar{a}_k)$, and $b_j = \bar{b}_j/(\sum_{k=1}^m \bar{b}_k)$. The cost matrix is set to $M_{ij} = (x_i - y_j)^2$.

We simulate different scales of the problems, $n = m = 1000, 5000$, and $10000$, and do not run Newton or APDAGD for $n \geq 5000$, as they are too time-consuming. The results are given in Figures 2 and 3.

As can be observed from the plots, the existing sparse Newton method SSNS performs well in Synthetic II but is slow in Synthetic I in terms of the run time. This is because the Hessian matrix is relatively dense in I, and SSNS needs to keep a large number of non-zero elements in the approximate Hessian, leading to slow linear system solving. SPLR, on the other hand, performs well in both settings, showing its adaptivity to different OT settings.

### 6.2. OT between a pair of vectorized images

We then study the OT problem between a pair of images. Specifically, we randomly select two images from the MNIST (Lecun et al., 1998) or Fashion-MNIST (Xiao et al., 2017) dataset, and let the $a$ and $b$ vectors be their flattened and normalized pixel values. The cost matrix holds the $\ell_1$-distances between individual pixels. These problems have a size of $n = m = 784$, with results shown in Figure 4 (for MNIST data) and Figure 5 (for Fashion-MNIST data), respectively.

The pattern in Figure 4 and Figure 5 shows that the Sinkhorn algorithm (*i.e.*, BCD) and the first-order method APDAGD have a quite slow convergence speed, and the sparse Newton method SSNS significantly accelerates the optimization. With this foundation, SPLR performs even better, as it combines the advantages of both sparse Newton methods and low-rank quasi-Newton methods.

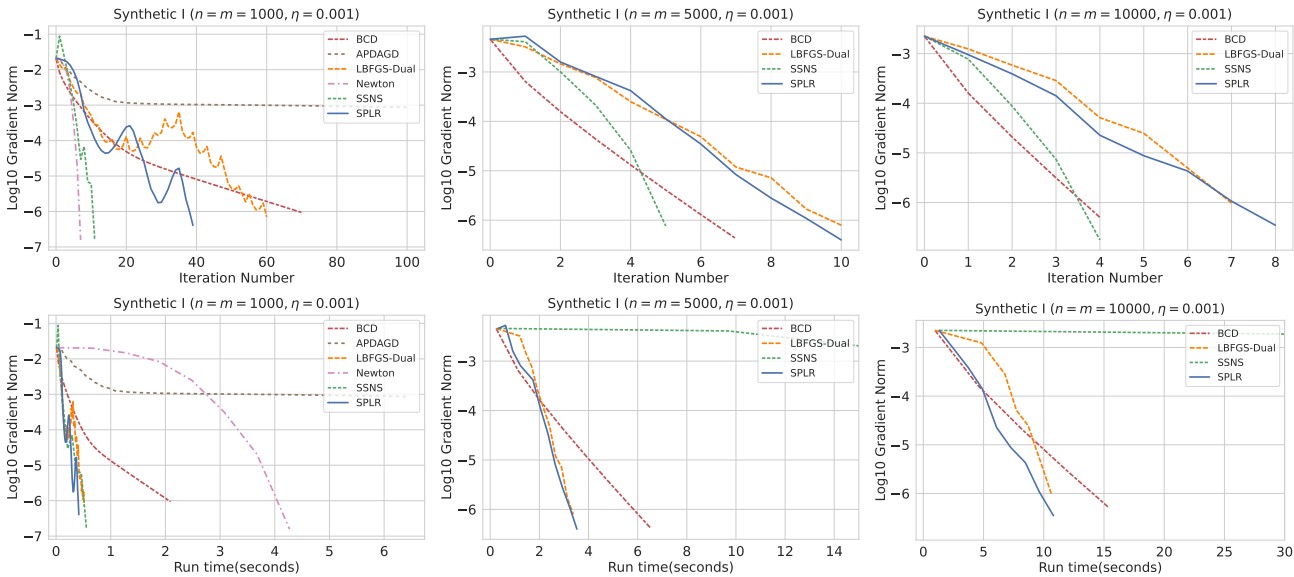

*Figure 2.* Performance of different algorithms on synthetic data I. Top: Gradient norm vs. iteration number for different problem sizes. Bottom: Gradient norm vs. run time.

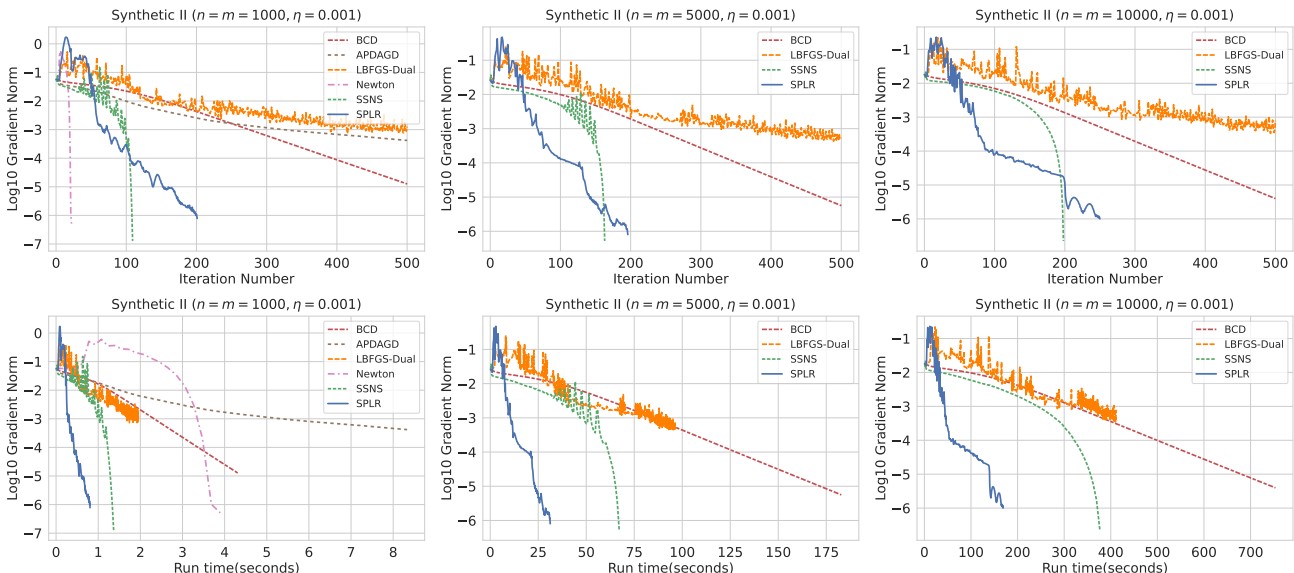

*Figure 3.* Performance of different algorithms on synthetic data II. Top: Gradient norm vs. iteration number for different problem sizes. Bottom: Gradient norm vs. run time.

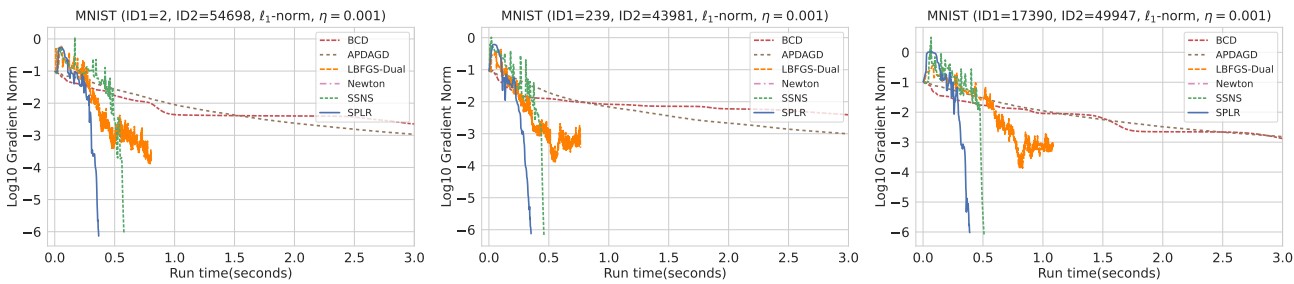

*Figure 4.* Performance of different algorithms on the MNIST data.

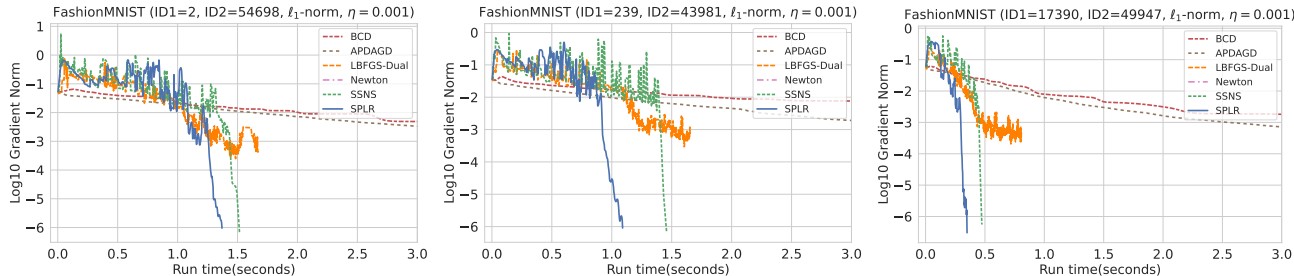

*Figure 5.* Performance of different algorithms on the Fashion-MNIST data.

## 6.3. OT between two classes of images

Finally, we reproduce the ImageNet experiment in Tang & Qiu (2024) that uses OT to characterize the difference between two data distributions. In particular, one class of images in the ImageNet dataset (Deng et al., 2009) is treated as one distribution, and each image in this class is an observation.

This experiment is interesting in that different values of the regularization parameter $\eta$ have a large impact on the performance of algorithms, as shown in Figure 6. With a small regularization, $\eta = 0.001$, BCD and APDAGD again have slow convergence speeds, and L-BFGS and SSNS perform quite well. The classic Newton method causes numerical issues, so it is not shown in the plot of $\eta = 0.001$. When $\eta$ is increased to 0.01, all methods converge faster except for SSNS. This is because a larger $\eta$ typically leads to a more dense Hessian matrix, so SSNS cannot use a sparse matrix to approximate the Hessian well. In contrast, SPLR deals with this situation well via its low-rank term, making SPLR perform consistently well on different $\eta$ values.

## 7. Conclusion

In this paper, we propose the SPLR quasi-Newton method to solve large-scale entropic-regularized OT, as a further extension of existing sparse Newton methods including the SNS (Tang et al., 2024) and SSNS (Tang & Qiu, 2024) methods. The design of the SPLR algorithm is highly dependent on the deepened theoretical understanding of the Hessian sparsification technique, which may be of interest by itself. On the other hand, the low-rank term introduced in SPLR effectively overcomes the limitation of sparse Newton methods in handling dense transport plans. In this sense, SPLR combines the best parts of purely low-rank-based methods (*e.g.*, L-BFGS) and purely sparsification-based methods (*e.g.*, SNS and SSNS). We anticipate that the technique developed in this paper would boost future exploration of highly efficient solvers for OT.

One potential future research direction is to study dimension-independent convergence rates of OT solvers. One known

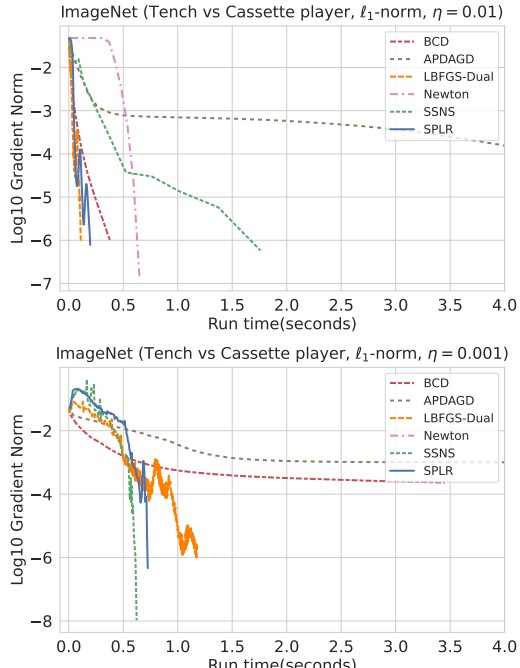

*Figure 6.* Performance of different algorithms on the ImageNet data. Top: $\eta = 0.01$. Bottom: $\eta = 0.001$.

result for the Sinkhorn algorithm is given in Carlier (2022), which shows that the Sinkhorn algorithm has a linear convergence rate that only depends on $\|M\|_\infty/\eta$ and not the dimension $(n, m)$. It is of interest to understand whether SPLR and other related solvers also have such properties.

## Impact Statement

This paper presents work whose goal is to advance the field of Machine Learning. There are many potential societal consequences of our work, none which we feel must be specifically highlighted here.

## Acknowledgements

Yixuan Qiu's work was supported in part by National Natural Science Foundation of China (12101389), Shanghai Pujiang Program (21PJC056), MOE Project of Key Research Institute of Humanities and Social Sciences (22JJD110001), and Shanghai Research Center for Data Science and Decision Technology.

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

# A. Additional Experiment Details

## A.1. Effect of low-rank terms

To verify that the low-rank term in the SPLR algorithm indeed plays a significant role in improving the sparse Newton method, we conduct an ablation study that compares SPLR with a purely sparsification-based method. We consider a special sparse Newton method that has the same sparsification scheme and hyperparameter setting as SPLR, and its only difference from SPLR is the lack of the low-rank term in the approximate Hessian. We examine the performance of the two methods on the synthetic data introduced in Section 6.1, with the results shown in Figures 7 and 8.

It is clear from the plots that in all the settings, SPLR converges faster than its counterpart, which highlights the benefits of the low-rank term.

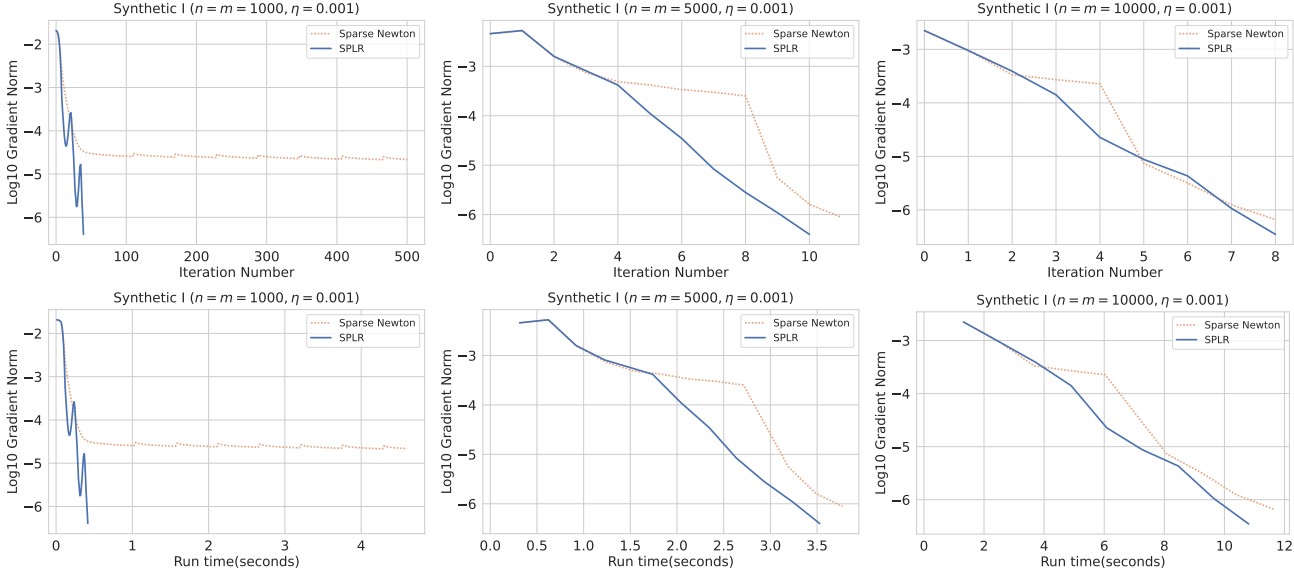

*Figure 7.* Comparing SPLR and sparse Newton method on synthetic data I. Top: Gradient norm vs. iteration number for different problem sizes. Bottom: Gradient norm vs. run time.

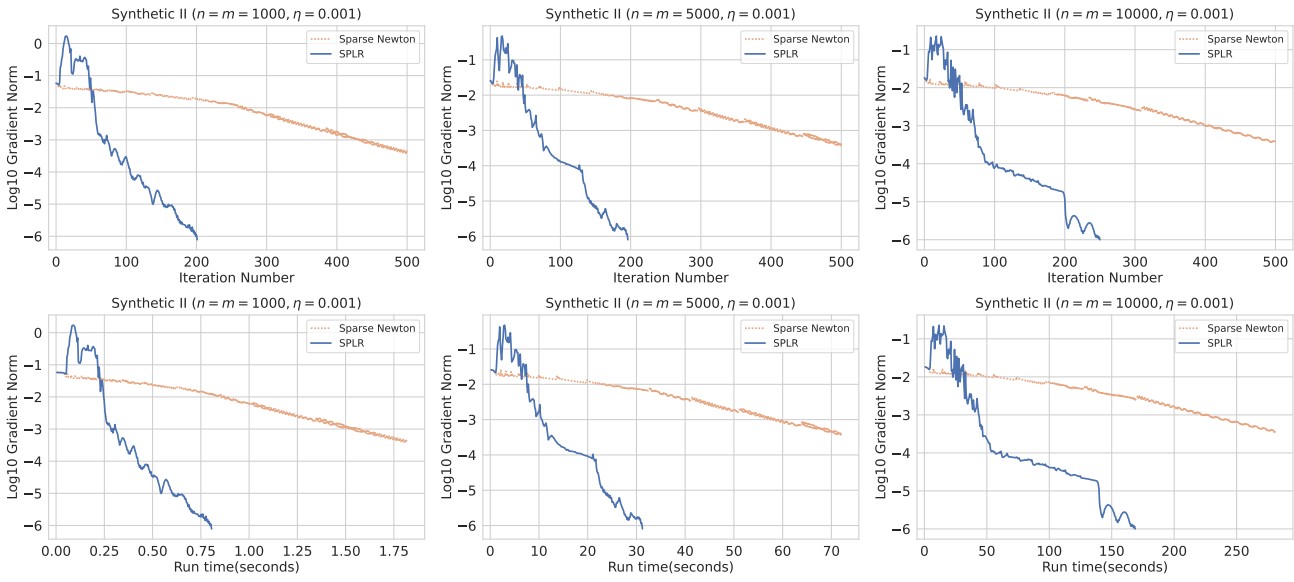

*Figure 8.* Comparing SPLR and sparse Newton method on synthetic data II.

## A.2. Effect of cost matrices

In this section, we include additional test examples in which the cost matrices are formed by the Euclidean distance instead of the $\ell_1$ distance. Figures 9 and 10 show the results on the (Fashion-)MNIST data and ImageNet data, respectively. Clearly, in all settings, SPLR is among the fastest solvers.

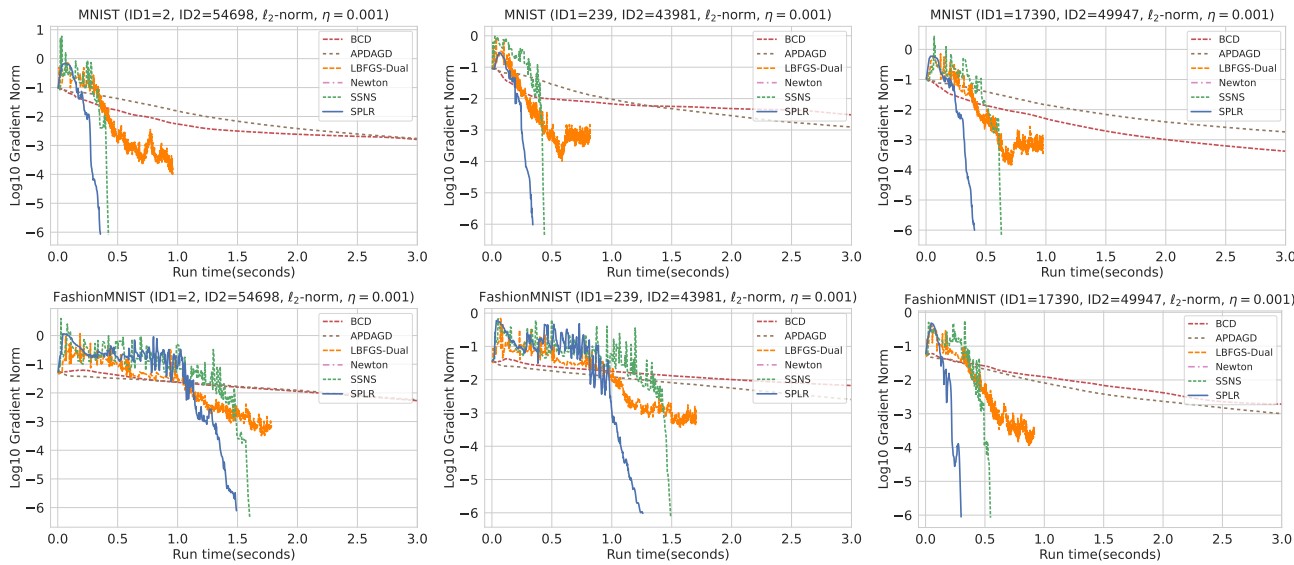

*Figure 9.* Performance of different algorithms on the MNIST (top row) and Fashion-MNIST (bottom row) data using the $\ell_2$-norm to form the cost matrices.

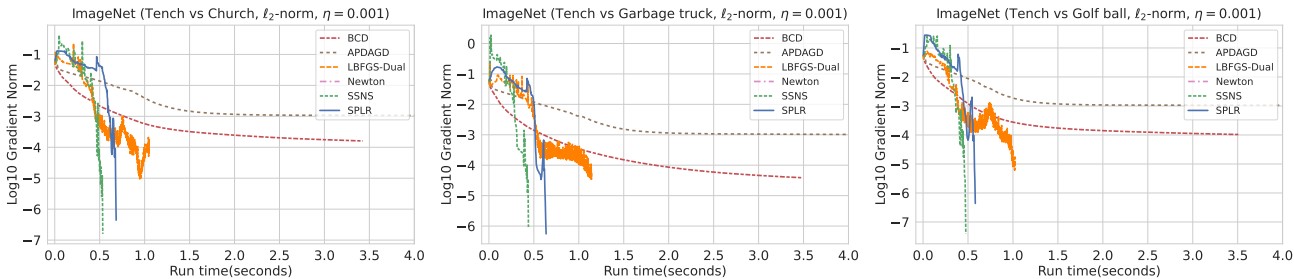

*Figure 10.* Performance of different algorithms on the ImageNet data using the $\ell_2$-norm to form the cost matrices.

## A.3. Effect of regularization parameters

Finally, we further validate the performance of the SPLR algorithm under the regularization parameter setting $\eta = 0.01$. Figures 11, 12, and 13 show the experiment results on the synthetic data, (Fashion-)MNIST data, and ImageNet data, respectively.

In summary, all these experiments show similar patterns to those in Section 6, validating the desirable performance of SPLR.

## A.4. Computing environment

All experiments in this article are conducted on a personal computer with an Intel i9-13900K CPU, 32 GB memory, and a Ubuntu 25.04 operating system.

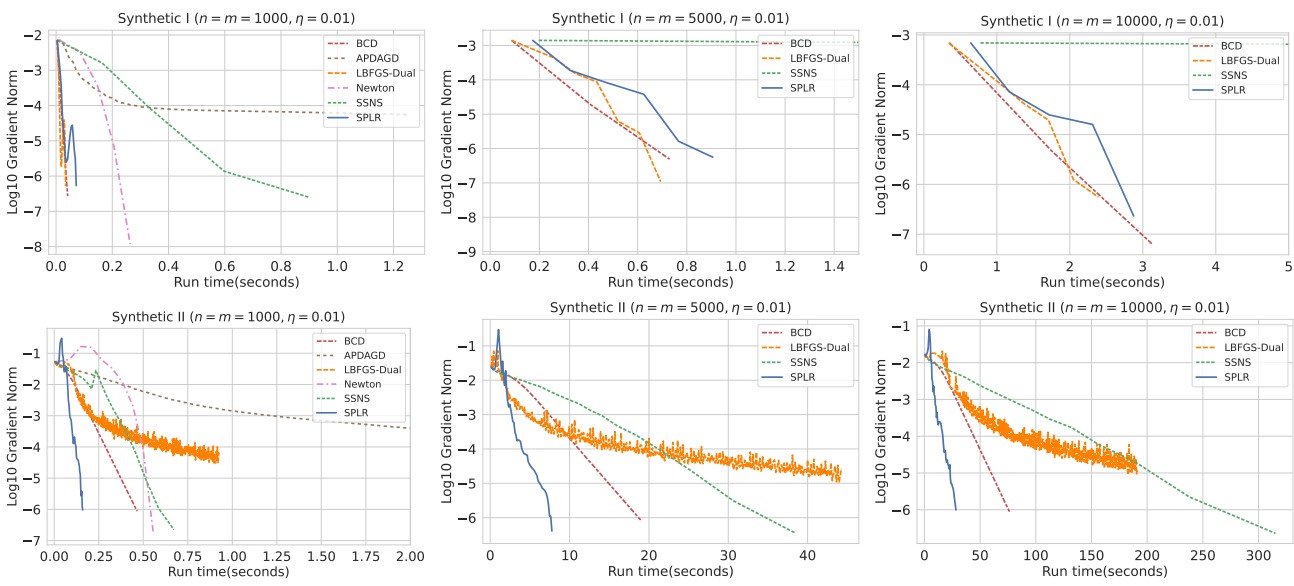

*Figure 11.* Performance of different algorithms on synthetic data I (top row) and synthetic data II (bottom row) with $\eta = 0.01$.

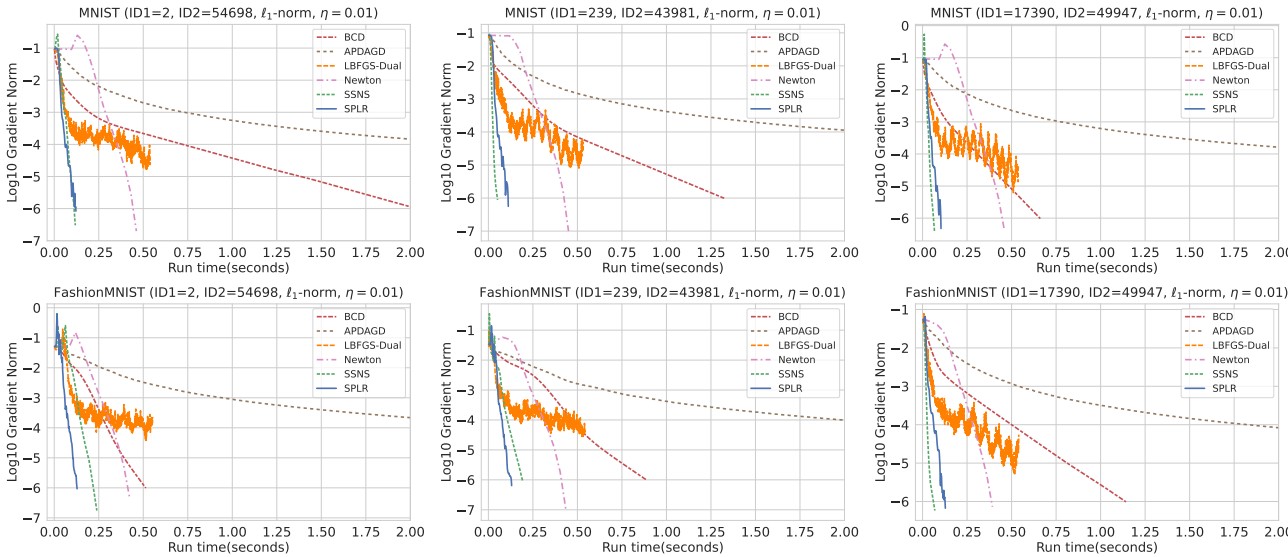

*Figure 12.* Performance of different algorithms on the MNIST (top row) and Fashion-MNIST (bottom row) data with $\eta = 0.01$.

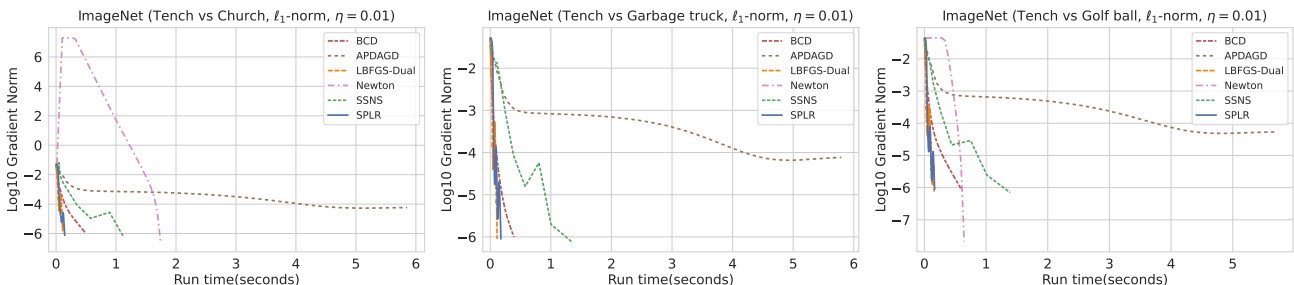

*Figure 13.* Performance of different algorithms on the ImageNet data with $\eta = 0.01$.

# B. Proofs of Theorems

## B.1. Technical Lemmas

**Lemma B.1.** *Let $M$ be a matrix of the form*

$$M = \begin{bmatrix} D_1 & R \\ R^T & D_2 \end{bmatrix},$$

*where $D_1 \in \mathbb{R}^{r \times r}$ and $D_2 \in \mathbb{R}^{s \times s}$ are diagonal matrices, and $R \in \mathbb{R}^{r \times s}$ has strictly positive entries. Suppose that $M$ is positive definite, and $M^{-1}$ has the partition*

$$M^{-1} = \begin{bmatrix} A_{r \times r} & B_{r \times s} \\ B^T & C_{s \times s} \end{bmatrix},$$

*and then we have $A > 0$, $C > 0$, and $B < 0$, where the inequality signs apply elementwisely.*

*Proof.* Since $M$ is positive definite, all of its diagonal elements must be strictly positive, implying that both $D_1$ and $D_2$ are invertible. By the inversion formula of block matrices, we have

$$A = (D_1 - R D_2^{-1} R^T)^{-1},$$
$$C = (D_2 - R^T D_1^{-1} R)^{-1},$$
$$B = -D_1^{-1} R (D_2 - R^T D_1^{-1} R)^{-1} = -D_1^{-1} R C.$$

Clearly, $J := D_1 - R D_2^{-1} R^T$ is the Schur complement of the block $D_2$ of the matrix $M$. By the properties of the Schur complement, we know that $J$ is positive definite, and hence it is nonsingular.

Moreover, since $R$ has strictly positive entries and $D_2$ has positive diagonal elements, we have that $R D_2^{-1} R^T$ has strictly positive entries. Therefore, $J$ can be represented in the form $J = sI - L$, where $L > 0$, $s \geq \rho(L)$, and $\rho(\cdot)$ stands for the spectral radius. Clearly, $J$ is irreducible, so by Theorem A(ii) of Meyer Jr & Stadelmaier (1978), $A = J^{-1}$ has strictly positive entries. The same argument can be used to prove that $C > 0$.

Finally, recall that $B = -D_1^{-1} R C$. Since $R > 0$, $C > 0$, and $D_1$ has positive diagonal elements, we conclude that $B < 0$.

$\square$

**Lemma B.2.** *Let $M$ be a matrix of the form*

$$M = \begin{bmatrix} A_{r \times r} & -B_{r \times s} \\ -B^T & C_{s \times s} \end{bmatrix},$$

*where $A > 0$, $B > 0$, $C > 0$, and the inequality signs apply elementwisely. Then $M$ has a positive eigenvalue $r$ such that any other eigenvalue of $M$ in absolute value is strictly smaller than $r$, and the eigenvector $v = (v_1^T, v_2^T)^T$ associated with $r$, where $v_1 \in \mathbb{R}^r$ and $v_2 \in \mathbb{R}^s$, can be normalized such that $v_1 > 0$ and $v_2 < 0$.*

*Proof.* Define

$$Q = \begin{bmatrix} I_r & O \\ O & -I_s \end{bmatrix},$$

and then it is easy to show that $Q^{-1} = Q$, and

$$QMQ^{-1} = \begin{bmatrix} I_r & O \\ O & -I_s \end{bmatrix} \begin{bmatrix} A & -B \\ -B^T & C \end{bmatrix} \begin{bmatrix} I_r & O \\ O & -I_s \end{bmatrix} = \begin{bmatrix} A & B \\ B^T & C \end{bmatrix} := \tilde{M}.$$

Therefore, $M$ and $\tilde{M}$ are similar to each other, and hence they must share the same eigenvalues. Clearly, $\tilde{M}$ is a positive matrix, so by the Perron–Frobenius theorem, it must have a positive and simple eigenvalue $r$ such that any other eigenvalue of $\tilde{M}$ in absolute value is strictly smaller than $r$. Moreover, $\tilde{M}$ has an eigenvector $\tilde{v} = (\tilde{v}_1^T, \tilde{v}_2^T)^T$ such that $\tilde{v}_1 \in \mathbb{R}^r$, $\tilde{v}_2 \in \mathbb{R}^s$, $\tilde{v} > 0$, and $\tilde{M}\tilde{v} = r\tilde{v}$.

Now let $v = Q^{-1}\tilde{v}$, and then

$$Mv = MQ^{-1}\tilde{v} = Q^{-1}QMQ^{-1}\tilde{v} = Q^{-1}\tilde{M}\tilde{v} = Q^{-1}r\tilde{v} = rv.$$

Therefore, $v$ is an eigenvector of $M$. Note that

$$v = Q^{-1}\tilde{v} = \begin{bmatrix} I_r & O \\ O & -I_s \end{bmatrix} \begin{bmatrix} \tilde{v}_1 \\ \tilde{v}_2 \end{bmatrix} = \begin{bmatrix} \tilde{v}_1 \\ -\tilde{v}_2 \end{bmatrix},$$

so $v_1 = \tilde{v}_1 > 0$ and $v_2 = -\tilde{v}_2 < 0$, which implies the stated result. $\square$

### B.2. Proof of Theorem 4.1

*Proof.* According to the sparsification scheme, the first row and first column of $\tilde{T}_{\Omega^*}$ is strictly greater than zero:

$$(\tilde{T}_{\Omega^*})_{1\cdot} > 0, \ (\tilde{T}_{\Omega^*})_{\cdot 1} > 0.$$

Suppose that $H_{\Omega^*}$ is of the form:

$$H_{\Omega^*} = \begin{bmatrix} A_{n\times n} & B_{n\times(m-1)} \\ C_{(m-1)\times n} & D_{(m-1)\times(m-1)} \end{bmatrix},$$

so in $H_{\Omega^*}$ we have:

$$A = \mathbf{diag}(A) > 0, \ D = \mathbf{diag}(D) > 0, \ B_{1\cdot} > 0, \ B_{\cdot 1} > 0, \ C_{1\cdot} > 0, \ C_{\cdot 1} > 0.$$

To show $H_{\Omega^*}^4 > 0$, we first condider elements of $H_{\Omega^*}^2$:

i. When $i \leq n, j \leq n$, $(H_{\Omega^*})_{ij}^2 = \langle (H_{\Omega^*})_{i\cdot}, (H_{\Omega^*})_{j\cdot} \rangle \geq B_{i,1}B_{j,1} > 0$.

ii. When $i > n, j > n$, $(H_{\Omega^*})_{ij}^2 = \langle (H_{\Omega^*})_{i\cdot}, (H_{\Omega^*})_{j\cdot} \rangle \geq C_{i-n,1}C_{j-n,1} > 0$.

iii. When $i \leq n, j = n + 1$, $(H_{\Omega^*})_{ij}^2 = \langle (H_{\Omega^*})_{i\cdot}, (H_{\Omega^*})_{j\cdot} \rangle \geq A_{i,i}C_{1,i} > 0$.

iv. When $i = n + 1, j \leq n$, $(H_{\Omega^*})_{ij}^2 = \langle (H_{\Omega^*})_{i\cdot}, (H_{\Omega^*})_{j\cdot} \rangle \geq A_{j,j}C_{1,j} > 0$.

v. When $i = 1, j > n$, $(H_{\Omega^*})_{ij}^2 = \langle (H_{\Omega^*})_{i\cdot}, (H_{\Omega^*})_{j\cdot} \rangle \geq B_{1,j-n}D_{j-n,j-n} > 0$.

vi. When $i > n, j = 1$, $(H_{\Omega^*})_{ij}^2 = \langle (H_{\Omega^*})_{i\cdot}, (H_{\Omega^*})_{j\cdot} \rangle \geq B_{1,i-n}D_{i-n,i-n} > 0$.

So we can assume that $H_{\Omega^*}^2$ is of the form:

$$H_{\Omega^*}^2 = \begin{bmatrix} A'_{n\times n} & B'_{n\times(m-1)} \\ C'_{(m-1)\times n} & D'_{(m-1)\times(m-1)} \end{bmatrix},$$

where

$$A' > 0, \ D' > 0, \ B'_{1\cdot} > 0, \ B'_{\cdot 1} > 0, \ C'_{1\cdot} > 0, \ C'_{\cdot 1} > 0.$$

Then consider elements of $H_{\Omega^*}^4$:

i. When $i \leq n, j \leq n$, $(H_{\Omega^*}^4)_{ij} = \langle (H_{\Omega^*}^2)_{i\cdot}, (H_{\Omega^*}^2)_{j\cdot} \rangle \geq A'_{i,1}A'_{j,1} > 0$.

ii. When $i > n, j > n$, $(H_{\Omega^*}^4)_{ij} = \langle (H_{\Omega^*}^2)_{i\cdot}, (H_{\Omega^*}^2)_{j\cdot} \rangle \geq D'_{1,i-n}D'_{1,j-n} > 0$.

iii. When $i \leq n, j > n$, $(H_{\Omega^*}^4)_{ij} = \langle (H_{\Omega^*}^2)_{i\cdot}, (H_{\Omega^*}^2)_{j\cdot} \rangle \geq C'_{1,i}D'_{1,j-n} > 0$.

iv. When $i > n, j \leq n$, $(H_{\Omega^*}^4)_{ij} = \langle (H_{\Omega^*}^2)_{i\cdot}, (H_{\Omega^*}^2)_{j\cdot} \rangle \geq C'_{1,j}D'_{1,i-n} > 0$.

Overall, we have $(H_{\Omega^*})^p > 0$ for $p = 4$, and hence $H_{\Omega^*}$ satisfies Assumption 3.2. □

**Lemma B.3.** *Given a sparsification scheme $\Omega$ that satisfies Assumption 3.2, the sparsified Hessian matrix $H_\Omega$ has the following properties:*

1. *The eigenvector $\mathbf{u}(H_\Omega)$ corresponding to $\lambda_{\max}(H_\Omega)$ can be normalized to have strictly positive entries;*

2. *There exists a neighborhood $N(H_\Omega)$ of $H_\Omega$ such that $\lambda_{\max}(H), H \in N(H_\Omega)$ is differentiable.*

*Proof.* Since $H_\Omega$ satisfies Assumption 3.2, there exists an integer $k > 0$ such that $H_\Omega^k > 0$. According to the Perron–Frobenius theorem, $\mathbf{u}(H_\Omega^k)$ can be normalized to have strictly positive entries and $\lambda_{\max}(H_\Omega^k)$ is strictly greater than all other eigenvalues of $H_\Omega^k$. Since $H_\Omega^k$ and $H_\Omega$ have the same eigenvectors, we can conclude that $\mathbf{u}(H_\Omega) = \mathbf{u}(H_\Omega^k)$ can be normalized to have strictly positive entries and the corresponding eigenvalue $\lambda_{\max}(H_\Omega)$ is strictly greater than all other eigenvalues of $H_\Omega$. This means that $\lambda_{\max}(H_\Omega)$ is a simple positive eigenvalue and therefore $\lambda_{\max}$ is differentiable at $H_\Omega$ according to Theorem 1 of Magnus (1985). □

**Lemma B.4.** *Given two sparsification schemes $\Omega_0$ and $\Omega_1$, suppose that $\Omega_1 \subseteq \Omega_0$. If $\Omega_1$ satisfies Assumption 3.2, then $\Omega_0$ also satisfies Assumption 3.2.*

*Proof.* Define $\Omega_\delta := \Omega_0 \backslash \Omega_1$, and $\Delta := \begin{bmatrix} O_{n \times n} & \tilde{T}_{\Omega_\delta} \\ \tilde{T}_{\Omega_\delta}^T & O_{(m-1) \times (m-1)} \end{bmatrix}$. Then we have

$$H_{\Omega_0} = H_{\Omega_1} + \Delta.$$

Since $\Omega_1$ satisfies Assumption 3.2, there must exist an integer $k > 0$ such that $(H_{\Omega_1})^k > 0$. Then for $H_{\Omega_0}$, we have:

$$(H_{\Omega_0})^k = (H_{\Omega_1} + \Delta)^k \geq (H_{\Omega_1})^k + \Delta^k \geq (H_{\Omega_1})^k > 0,$$

where all inequality signs are elementwise. This means that $H_{\Omega_0}$ also satisfies Assumption 3.2. □

### B.3. Proof of Theorem 3.3

*Proof.* First, we prove $\lambda_{\max}(H_{\Omega_0}) > \lambda_{\max}(H_{\Omega_1})$.

Define $p = i, q = n + j, \beta = (H_{\Omega_0})_{pq}$, then the difference of Hessian is:

$$H_{\Omega_1} - H_{\Omega_0} = -\beta \left( \mathbf{e}_p \mathbf{e}_q^T + \mathbf{e}_q \mathbf{e}_p^T \right) := -\beta \mathbf{J},$$

Define $M(\kappa) = H_{\Omega_0} - \kappa J, l(\kappa) = \lambda_{\max}(M(\kappa)), \kappa \in \mathbb{R}$, then we have:

$$\lambda_{\max}(H_{\Omega_1}) - \lambda_{\max}(H_{\Omega_0}) = l(\beta) - l(0),$$

Since $H_{\Omega_1}$ satisfies Assumption 3.2, we can show that $\lambda_{\max}$ is differentiable on $\{M(\kappa)|\kappa \in [0, \beta]\}$ according to Lemma B.3. Suppose the eigenvector associated with $\lambda_{\max}(M(\kappa))$ is $\mathbf{u}_\kappa$, then the derivative at $M(\kappa)$ is:

$$\left. \frac{\partial \lambda_{\max}}{\partial M} \right|_{M=M(\kappa)} = \mathbf{u}_\kappa \mathbf{u}_\kappa^T$$

thus $l'(\kappa) = \langle \mathbf{u}_\kappa \mathbf{u}_\kappa^T, -J \rangle$. According to the Lagrange's mean value theorem, $\exists \xi \in (0, \beta)$ such that:

$$\begin{aligned} l(\beta) - l(0) &= l'(\xi)(\beta - 0) \\ &= \langle \mathbf{u}_\xi \mathbf{u}_\xi^T, -J \rangle \beta \\ &= -\beta \left[ tr(\mathbf{u}_\xi \mathbf{u}_\xi^T \mathbf{e}_p \mathbf{e}_q^T) + tr(\mathbf{u}_\xi \mathbf{u}_\xi^T \mathbf{e}_q \mathbf{e}_p^T) \right] \\ &= -\beta \left[ tr(\mathbf{u}_\xi^T \mathbf{e}_p \mathbf{e}_q^T \mathbf{u}_\xi) + tr(\mathbf{u}_\xi^T \mathbf{e}_q \mathbf{e}_p^T \mathbf{u}_\xi) \right] \\ &= -2\beta (\mathbf{u}_\xi)_p (\mathbf{u}_\xi)_q \end{aligned} \tag{10}$$

According to Lemma B.3, $\mathbf{u}_\xi$ can be normalized to have strictly positive entries so that $(\mathbf{u}_\xi)_p(\mathbf{u}_\xi)_q > 0$, which means $l(\beta) - l(0) = \lambda_{\max}(H_{\Omega_1}) - \lambda_{\max}(H_{\Omega_0}) < 0$.

Then we prove $\lambda_{\min}(H_{\Omega_0}) < \lambda_{\min}(H_{\Omega_1})$.

According to Tang & Qiu (2024), we have $M(\kappa) \succ 0$ thus invertible, then according to Lemma B.1, $M(\kappa)^{-1}$ is of the form:

$$M(\kappa)^{-1} = \begin{pmatrix} A & B \\ B^T & C \end{pmatrix}$$

where $A > 0, C > 0, B < 0$. Then according to Lemma B.2, we can show that $\lambda_{\max}(M(\kappa)^{-1})$ is a simple positive eigenvalue, the corresponding eigenvector $\mathbf{v}_\kappa = (\mathbf{v}_1^T, \mathbf{v}_2^T)^T$ where $\mathbf{v}_1 \in \mathbb{R}^n, \mathbf{v}_2 \in \mathbb{R}^{m-1}$ can be normalized such that $\mathbf{v}_1 > 0, \mathbf{v}_2 < 0$. Since $\mathbf{v}_\kappa$ is also the eigenvector of $M(\kappa)$ corresponding to $\lambda_{\min}(M(\kappa))$, we can tell that $(\mathbf{v}_\kappa)_p(\mathbf{v}_\kappa)_q < 0$, where $p \leq n, q > n$.

Define $h(\kappa) = \lambda_{\min}(M(\kappa))$, then similar to (10) we have $h(\beta) - h(0) = -2\beta(\mathbf{v}_\xi)_p(\mathbf{v}_\xi)_q$, where $\xi \in (0, \beta)$. Recall that $(\mathbf{v}_\xi)_p(\mathbf{v}_\xi)_q < 0$, so $h(\beta) - h(0) > 0$, which means $\lambda_{\min}(H_{\Omega_0}) < \lambda_{\min}(H_{\Omega_1})$. $\qquad\square$

### B.4. Proof of Theorem 5.1

By definition,
$$B_{k+1} = H_\Omega^{k+1} + auu^T + bvv^T + \tau_{k+1}I,$$

where
$$u = y_k, \ v = (H_\Omega^{k+1} + \tau_{k+1})s_k, \ a = \frac{1}{y_k^T s_k}, \ b = -\frac{1}{s_k^T(H_\Omega^{k+1} + \tau_{k+1})s_k}.$$

By the design of the algorithm, we have $y_k^T s_k > 0$, so $auu^T$ and $bvv^T$ are rank-one matrices with $a > 0$ and $b < 0$. Then we have
$$\lambda_{\max}(auu^T) = au^Tu, \quad \lambda_{\max}(bvv^T) = 0.$$

It is well known that $\lambda_{\max}(A + B) \leq \lambda_{\max}(A) + \lambda_{\max}(B)$ for symmetric matrices $A$ and $B$, so

$$\lambda_{\max}(B_{k+1}) \leq \lambda_{\max}(H_\Omega^{k+1}) + \frac{y_k^T y_k}{y_k^T s_k} + \tau_{k+1}.$$

Corollary 3.4 shows that $\lambda_{\max}(H_\Omega^{k+1}) \leq \lambda_{\max}(H_{k+1})$, and then by the assumption of the theorem, we have

$$\lambda_{\max}(H_\Omega^{k+1}) \leq \lambda_{\max}(H_{k+1}) \leq U.$$

On the other hand, since $f(x)$ is twice differentiable, the mean value theorem indicates that

$$y_k = g(x_{k+1}) - g(x_k) = \bar{G}_k(x_{k+1} - x_k) = \bar{G}_k s_k,$$

where
$$\bar{G}_k = \int_0^1 H((1-t)x_k + tx_{k+1})\mathrm{d}t.$$

Again by the assumption, $(1-t)x_k + tx_{k+1} \in D$ for all $t \in [0,1]$, so for any $v \in \mathbb{R}^{n+m-1}$,

$$v^T\bar{G}_k v = \int_0^1 v^T H((1-\tau)x_k + \tau x_{k+1})v\mathrm{d}\tau \geq \int_0^1 Lv^T v\mathrm{d}\tau = Lv^T v,$$

and similarly, $v^T\bar{G}_k v \leq Uv^T v$. This indicates that

$$0 < L \leq \lambda_{\min}(\bar{G}_k) \leq \lambda_{\max}(\bar{G}_k) \leq U.$$

As a result,
$$au^Tu = \frac{y_k^T y_k}{y_k^T s_k} = \frac{s_k^T \bar{G}_k^2 s_k}{s_k^T \bar{G}_k s_k} = \frac{w^T \bar{G}_k w}{w^T w},$$

where $w = (\bar{G}_k)^{1/2} s_k$ is well-defined since $\bar{G}_k$ is positive definite. Then by the properties of eigenvalues, we get $a u^T u \leq \lambda_{\max}(\bar{G}_k) \leq U$. Overall, we have

$$\lambda_{\max}(B_{k+1}) \leq \lambda_{\max}(H_\Omega^{k+1}) + \frac{y_k^T y_k}{y_k^T s_k} + \tau_{k+1} \leq 2U + \tau_{\max}.$$

Now consider the inverse of $B_{k+1}$. Using the Sherman–Morrison–Woodbury formula, we can obtain

$$B_{k+1}^{-1} = U^T (H_\Omega^{k+1} + \tau_{k+1} I)^{-1} U + \frac{1}{y_k^T s_k} \cdot s_k s_k^T,$$

where $U = I - (y_k^T s_k)^{-1} y_k s_k^T$. Let $\bar{H}_{k+1} = H_\Omega^{k+1} + \tau_{k+1} I$, and then $\bar{H}_{k+1}$ is positive definite as $H_\Omega^{k+1}$ is positive definite. So for any vector $v \in \mathbb{R}^{n+m-1}$,

$$v^T U^T \bar{H}_{k+1}^{-1} U v \leq \lambda_{\max}(\bar{H}_{k+1}^{-1}) \|Uv\|^2 = \frac{v^T U^T U v}{\lambda_{\min}(\bar{H}_{k+1})} \leq \frac{\lambda_{\max}(U^T U)}{\lambda_{\min}(\bar{H}_{k+1})} \cdot \|v\|^2.$$

Note that

$$U^T U = I - \frac{1}{y_k^T s_k}(y_k s_k^T + s_k y_k^T) + \frac{y_k^T y_k}{(y_k^T s_k)^2} \cdot s_k s_k^T.$$

Since $U^T U$ is positive semi-definite, we have $\lambda_{\max}(U^T U) = \|U^T U\|$, where $\|\cdot\|$ represents the operator norm for matrices. Therefore,

$$
\begin{aligned}
\left\| \frac{1}{y_k^T s_k}(y_k s_k^T + s_k y_k^T) \right\| &\leq \frac{2\|y_k s_k^T\|}{y_k^T s_k} = \frac{2\|\bar{G}_k s_k s_k^T\|}{s_k^T \bar{G}_k s_k} \\
&\leq \frac{2\|\bar{G}_k\| \cdot \|s_k s_k^T\|}{s_k^T \bar{G}_k s_k} = \frac{2\|\bar{G}_k\| \cdot s_k^T s_k}{s_k^T \bar{G}_k s_k} \\
&\leq \frac{2U}{L}.
\end{aligned}
$$

Similarly,

$$\left\| \frac{y_k^T y_k}{(y_k^T s_k)^2} \cdot s_k s_k^T \right\| = \frac{y_k^T y_k}{y_k^T s_k} \cdot \frac{s_k^T s_k}{s_k^T \bar{G}_k s_k} \leq \frac{U}{L}.$$

So overall, $\|U^T U\| \leq 1 + 3U/L$, and then

$$\lambda_{\max}\left(U^T \bar{H}_{k+1}^{-1} U\right) \leq \frac{\lambda_{\max}(U^T U)}{\lambda_{\min}(\bar{H}_{k+1})} \leq \frac{1}{L}\left(1 + \frac{3U}{L}\right).$$

Finally, we obtain

$$\lambda_{\max}(B_{k+1}^{-1}) \leq \frac{1}{L}\left(1 + \frac{3U}{L}\right) + \frac{s_k^T s_k}{y_k^T s_k} \leq \frac{1}{L}\left(2 + \frac{3U}{L}\right).$$

Clearly, $B_{k+1}^{-1}$ is positive semi-definite, so

$$\lambda_{\min}(B_{k+1}) = \frac{1}{\lambda_{\max}(B_{k+1}^{-1})} \geq \left(2 + \frac{3U}{L}\right)^{-1} L.$$

### B.5. Proof of Corollary 5.2

Consider the level set $D = \{x : f(x) \leq f(x_0)\}$. Clearly, $D$ is a closed convex set, and there exist constants $L, U > 0$ such that $\lambda_{\min}(H(x)) \geq L$ and $\lambda_{\max}(H(x)) \leq U$ for all $x \in D$.

Let $\theta_k$ be the angle between $-g_k$ and the search direction $p_k = -B_k^{-1} g_k$. Then clearly,

$$\cos\theta_k = \frac{-g_k^T p_k}{\|g_k\| \cdot \|p_k\|} = \frac{p_k^T B_k p_k}{\|B_k p_k\| \cdot \|p_k\|} \geq \frac{\lambda_{\min}(B_k)\|p_k\|^2}{\lambda_{\max}(B_k)\|p_k\|^2} = \frac{\lambda_{\min}(B_k)}{\lambda_{\max}(B_k)}. \tag{11}$$

Then by Theorem 5.1, we have

$$\cos \theta_k \geq \frac{\lambda_{\min}(B_k)}{\lambda_{\max}(B_k)} \geq c := \left(2 + \frac{3U}{L}\right)^{-1} \frac{L}{2U + \tau_{\max}} > 0.$$

Zoutendijk's theorem (see for example Theorem 3.2 of Nocedal & Wright, 2006) shows that

$$\sum_{k \geq 0} (\cos \theta_k)^2 \|g_k\|^2 < \infty,$$

so we must have

$$c^2 \sum_{k \geq 0} \|g_k\|^2 < \infty,$$

which implies that $\|g_k\| \to 0$ as $k \to \infty$.

### B.6. Proof of Theorem 5.3

Similar to the proof of Corollary 5.2, consider the level set $D = \{x : f(x) \leq f(x_0)\}$, and we have $\lambda_{\min}(H(x)) \geq L$ and $\lambda_{\max}(H(x)) \leq U$ for all $x \in D$.

Lemma 2.1 of Byrd et al. (1987) shows that for any $k \geq 1$,

$$f(x_{k+1}) - f^* \leq [1 - Lc_1 \tilde{c}_2 \cos^2 \theta_k] \cdot [f(x_k) - f^*],$$

where $\tilde{c}_2 = (1 - c_2)/U$, and $\cos \theta_k$ is defined in (11). Take

$$r = 1 - Lc_1 \tilde{c}_2 c^2, \quad c = \left(2 + \frac{3U}{L}\right)^{-1} \frac{L}{2U + \tau_{\max}},$$

and then we get the desired result.

