# OpenReview forum: "The Sparse-Plus-Low-Rank Quasi-Newton Method for Entropic-Regularized Optimal Transport"
_ICML.cc/2025/Conference — ICML 2025 poster_

### Official Review · Reviewer_aArm · 2025-03-06

**Overall Recommendation:** 2

**Summary:**

This paper is about solving entropic-regularized optimal transport problem via a quasi-Newton method. Entropic-regularized OT problem has applications in machine learning, but its solution is difficult to find. The paper first presents some theoretical analysis Hessian sparsification, which can be used to solve the OT problem in a more tractable way. The paper then presents a low-rank approximation to the Hessian matrix to facilitate computations. The paper later gives theoretical analysis of the method and validates its performance via experiments.

**Claims And Evidence:**

Yes

**Essential References Not Discussed:**

No

**Experimental Designs Or Analyses:**

Not always, as the paper barely touches on machine learning.

**Methods And Evaluation Criteria:**

Yes

**Other Comments Or Suggestions:**

Please see strengths and weaknesses.

**Other Strengths And Weaknesses:**

The strength includes a comprehensive analysis of using sparse Hessian and a low-rank approximation in solving entropic-regularized OT problem and a large amount of experiments. Another strength is to set the \tau dynamically so when the solution is close to the minimum, \tau is updated more gradually.

Biggest weakness is the seemingly lack of relevance of this paper to machine learning. While OT is indeed widely used in machine learning, this paper barely covers any machine learning techniques or applications. This makes the paper somehow out of scope of this conference.

In Eq. (5), is the low-rank approximation always needed? If not, when is it needed?

In Eq. (6) and (7), how is u, actually u_k, determined? It seems it was never defined in the main text, though it was given in the supplementary material, and it seems u_k was a typo. And can the authors give some explanation about the meaning of a, b, u, v in Eq. (6) and (7)? That will help readers see the motivation behind the method.


The experimental results are not always clear to me. In Figure 1, especially second row, it seems the proposed SPLR method had similar run-time vs log10 Gradient norm curves as several existing methods.

In Figure 2, when the Hessian is not so dense, the advantage of SPLR method is more obvious. Considering Figure 1 and Figure 2 together, it seems to indicate the SPLR method works better if Hessian is not so dense in the first place.

Figure 4 seems to indicate that the regularization weight is critical in determining each method's speed of convergence. While SPLR gave better performance than some methods, it converged slower than a few other methods. If so, that means a careful selection of the regularization weight \eta is necessary, and this may increase the computational load of the proposed method (as well as the other methods). Figure 11 shows similar patterns.

Another concern is that it seems SPLR experienced large oscillation in reducing Log10 Gradient Norm, though some other methods had the similar behavior, but this may raise the risk of early or incorrect stopping of iterations. Can authors elaborate on this?

**Questions For Authors:**

Please see strengths and weaknesses.

**Relation To Broader Scientific Literature:**

It is related to relevant literature in using OT in machine learning.

**Theoretical Claims:**

Yes

---

> ### Author Rebuttal · Authors · 2025-04-01
>
> Thank you for your detailed review and thoughtful comments. Please see our point-by-point responpses below.
>
> ---
>
> > Not always, as ... on machine learning.
>
> > Biggest weakness is ... of this conference.
>
> We sincerely appreciate your feedback on our manuscript. Your observation regarding the limited discussion on machine learning is invaluable and has encouraged us to further explore this perspective to enhance our work.
>
> To provide a more comprehensive overview, we have reviewed recent advancements in computational optimal transport (OT) within the machine learning community, particularly those presented at ICML. Our findings indicate a substantial body of literature [1-4] that focuses on this topic, underscoring its relevance within the field.
>
> Furthermore, the [ICML 2025 Call for Papers webpage](https://icml.cc/Conferences/2025/CallForPapers) includes "Optimization (convex and non-convex optimization, matrix/tensor methods, stochastic, online, non-smooth, composite, etc.)" as a topic of interest, highlighting the strong connection between our research and ongoing discussions in the machine learning community.
>
> Once again, we sincerely appreciate your insights, and we would add more discussions on the importance of OT optimization in the revised paper.
>
> **References**
>
> [1] Dvurechensky, Pavel, Alexander Gasnikov, and Alexey Kroshnin. "Computational optimal transport: Complexity by accelerated gradient descent is better than by Sinkhorn’s algorithm." ICML, 2018.
>
> [2] Lu, Haihao, Robert Freund, and Vahab Mirrokni. "Accelerating greedy coordinate descent methods." ICML, 2018.
>
> [3] Lin, Tianyi, Nhat Ho, and Michael Jordan. "On efficient optimal transport: An analysis of greedy and accelerated mirror descent algorithms." ICML, 2019.
>
> [4] Guminov, Sergey, et al. "On a combination of alternating minimization and Nesterov’s momentum." ICML, 2021.
>
> > In Eq. (5), ... is it needed?
>
> Thank you for your insightful question. We cannot guarantee that the low-rank approximation is always necessary. However, based on our numerical experiments, particularly in Figures 5 and 6, we have observed that adding the low-rank term generally leads to faster convergence of the algorithm. Specifically, when the curvature condition $(s^k)^T y^k > \varepsilon \|y^k\|^2$ holds, we consistently find it beneficial to include the low-rank term.
>
> > In Eq. (6) ... behind the method.
>
> Thank you for pointing this out! The correct notation should be $u = y_k$, and we will fix it in the main text.
>
> As for the variables $(a, b, u, v)$ in Eq. (6) and (7), they are mostly notational: $u$ and $v$ represent two low-rank components, derived from the BFGS quasi-Newton method, and $a$ and $b$ are their respective scaling coefficients, determined by the secant equation (Eq. 6).
>
> These terms collectively help ensure a more stable and efficient update, improving the convergence behavior of the method. We will expand the explanation in the main text to better highlight the motivation behind these choices.
>
> > The experimental results ... in the first place.
>
> Thank you for your insightful observation. Indeed, our results confirm that the SPLR method is as good as alternatives in virtually all scenarios, and significantly outperforms others when the Hessian is sparse. This demonstrates the robustness of SPLR in worst cases and its high efficiency in favorable cases.
>
> This behavior is due to the sparse-plus-low-rank design, which ensures stable and efficient performance across different Hessian structures.
>
> > Figure 4 seems to ... shows similar patterns.
>
> We appreciate your observation about the impact of the regularization weight $\eta$ on convergence speed. Indeed, as shown in Figures 4 and 11, the choice of $\eta$ significantly affects all methods, including ours. While our paper focuses on optimizing the OT problem *given* a fixed $\eta$, we acknowledge that selecting $\eta$ is critical in practice—though this lies outside our current scope. Investigating systematic strategies for choosing $\eta$ efficiently could be an interesting direction for future work.
>
> > Another concern is ... elaborate on this?
>
> Thank you for your valuable observation. We appreciate your concern regarding the oscillation in the Log10 Gradient Norm observed in SPLR. However, we would like to clarify that such oscillations do not pose a risk of early or incorrect stopping. In fact, the gradient norm serves as a rigorous stopping criterion. Specifically, when the gradient norm is reduced below a predefined threshold, we can theoretically prove that the optimality gap is also bounded by a corresponding threshold. This ensures that the stopping condition is reliable and provides confidence in the convergence of the method.
>
> On the other hand, our method guarantees that the dual objective function value is nonincreasing across iterations, so if the stopping criterion is on the function value, it will not have any oscillation.

---

### Official Review · Reviewer_Ybnz · 2025-03-12

**Overall Recommendation:** 2

**Summary:**

This paper proposes a Sparse-Plus-Low-Rank Quasi-Newton （SPLR）method for entropic regularized Optimal Transport (OT). The proposed algorithm improves the approximation of the Hessian matrix by adding a low-rank term, thus better solving the dense situation, effectively solving the entropic-regularized OT problem and additionally reducing the amount of computation. The theoretical analysis, experimental validation, and convergence guarantee in the paper are all  sufficient, demonstrating the superior performance of the SPLR algorithm.

**Claims And Evidence:**

Yes

**Essential References Not Discussed:**

No

**Experimental Designs Or Analyses:**

Yes

**Methods And Evaluation Criteria:**

Yes

**Other Comments Or Suggestions:**

(1) It is mentioned that the algorithm has a lower time complexity, but it relies on the choice of sparse matrices during the computation process, and the paper does not provide specific theoretical analysis.

(2) The paper mentions that during the experimental process, the algorithm can achieve super-linear-like convergence speed, but in the theoretical proof, the algorithm only reaches linear convergence speed. What is the cause of this phenomenon? Is it possible to further derive corresponding results in theory?

(3) Although the paper provides a brief introduction to existing quasi-Newton algorithms, it does not compare their convergence rates with those of classical algorithms. It is recommended to add a comparison table to illustrate the superiority of the algorithm.

(4) How is the boundedness of H(x) ensured during the iteration process?

(5) In the article, a rank-2 approximation term was added to achieve a faster convergence rate. Can this approximation term be of a higher rank?

(6) Is there an error in line 16 of Algorithm 1?

**Other Strengths And Weaknesses:**

**Strengths: **

(1) The authors effectively enhanced the algorithm's efficiency by integrating a low-rank matrix into the sparse approximation framework for Hessian, with rigorous theoretical analysis demonstrating both convergence properties and computational complexity reduction, well-supported by theoretical analysis.

(2) The empirical tests conducted on both synthetic and actual datasets, highlight the practical superiority of SPLR compared to conventional techniques such as Sinkhorn, L-BFGS, and SSNS. The ablation study, detailed in Appendix A.1, convincingly confirms the essential role of the low-rank component.

**Weaknesses:**

(1) It is mentioned that the algorithm has a lower time complexity, but it relies on the choice of sparse matrices during the computation process, and the paper does not provide specific theoretical analysis.

(2) The paper mentions that during the experimental process, the algorithm can achieve super-linear-like convergence speed, but in the theoretical proof, the algorithm only reaches linear convergence speed. What is the cause of this phenomenon? Is it possible to further derive corresponding results in theory?

**Questions For Authors:**

Please see the above comments and suggestions.

**Relation To Broader Scientific Literature:**

Quasi-Newton Method

**Theoretical Claims:**

Yes

---

> ### Author Rebuttal · Authors · 2025-04-01
>
> Thank you for your detailed review and thoughtful comments. Please see our point-by-point responpses below.
>
> ---
>
> ### Weakness 1/Comment 1
>
> We thank you for raising this important point about our complexity analysis. Let us clarify the theoretical aspects of our algorithm's computational efficiency:
>
> 1. **Theoretical per-iteration cost**: As noted in Section 5 (Lines 267-270), each iteration's dominant cost is the inversion $(H_{\Omega}^{k+1} + \tau_{k+1} I)^{-1}$ in the computation of $B_{k+1}^{-1}$. According to Chapter 10 in (Shewchuk, J. R., 1994), when it is solved using conjugate gradient method, the time complexity is $O(\rho n m \sqrt{\kappa})$, where $\kappa$ is the condition number of $H_{\Omega}^{k+1}$.
> 2. **Convergence Guarantees**:
>    - Theorem 5.1 establishes linear convergence
>    - The convergence constant depends on $L$ and $U$
>
> You are absolutely correct that the sparse pattern selection affects complexity, and we appreciate the opportunity to clarify this relationship. Our method provides explicit control over this through $\rho$ while maintaining theoretical convergence guarantees.
>
> ### Weakness 2/Comment 2
>
> We sincerely thank you for identifying this important gap between our theoretical and empirical convergence results. The observed superlinear-like behavior indeed requires deeper investigation, and we offer the following possible causes:
>
> - *Approximation quality improvement*: As optimization progresses (especially near the solution), our SPLR approximation appears to capture increasingly accurate Hessian information. This is evidenced in Fig. 2 (columns 2-3) where the log gradient norm decreases sharply after certain iterations.
>
> - *Hybrid behavior*: The method may initially follow linear convergence (as proved) but transition to superlinear once the approximation error becomes sufficiently small.
>
> We greatly appreciate your suggestion to pursue this theoretically, as it could lead to significant new insights about Hessian approximation methods. The gap between our current theory and observations points to exciting future research directions.
>
> ### Comment 3
>
> We thank you for this valuable suggestion. Below is a proposed comparison table we will include in the revised manuscript to clearly demonstrate the advantages of our SPLR method:
>
> | Method          | Convergence Rate                                 | Per-Iteration Cost                                                          | Memory Usage               | Hessian Approximation Type |
> | --------------- | ------------------------------------------------ | --------------------------------------------------------------------------- | -------------------------- | -------------------------- |
> | Newton's Method | Quadratic                                        | $O(n^3)$                                                                    | $O(n^2)$                   | Exact                      |
> | BFGS            | Superlinear                                      | $O(n^2)$                                                                    | $O(n^2)$                   | Dense low-rank update      |
> | L-BFGS          | Linear                                           | $O(nm)$ ($m$: memory)                                                       | $O(nm)$                    | Limited-memory low-rank    |
> | SNS<br>         | Superlinear (conjectured)                        | $O(\rho(\varepsilon) n^2 \sqrt{\kappa})$ ($\varepsilon$: element threshold) | $O(\rho(\varepsilon) n^2)$ | Sparse only                |
> | SSNS            | Quadratic                                        | $O(\rho(\delta) n^2 \sqrt{\kappa})$ ($\delta$: error threshold)        | $O(\rho(\delta)n^2)$       | Sparse only                |
> | SPLR            | Theoretically linear;<br>emperically superlinear | $O(\rho n^2 \sqrt{\kappa})$                                                 | $O(\rho n^2)$              | Sparse + Low-rank          |
>
> Your suggestion significantly improves our paper's ability to communicate the method's advantages, and we appreciate this constructive feedback.
>
> ### Comment 4
>
> Thanks for raising this question. The basic idea is as follows: since our method guarantees that the objective function value is nonincreasing, the iterates $\{x_k\}$ are restricted to the level set $D = \{x: f(x) \leq f(x_0)\}$, which can be proved to be compact. Since $H(x)$ is continuous on $x$, it must be bounded on $D$. Of course, the bounds may depend on the initial value $x_0$.
>
> ### Comment 5
>
> We thank you for this insightful technical question. Our rank-2 approximation is motivated by the rank-2 modification scheme in the BFGS quasi-Newton method, and can indeed be naturally generalized to higher ranks with all theoretical gurantees. The question highlights an important implementation flexibility that deserves more explicit treatment and we will adjust our manualscript accordingly.
>
> ### Comment 6
>
> We thank you for pointing out this typo. It should be $auu^T + bvv^T$.

---

### Official Review · Reviewer_pXWg · 2025-03-14

**Overall Recommendation:** 4

**Summary:**

The paper concerns faster solver for the optimal transport (OT) problem, by proposing a new type of Hessian approximation within the quasi-Newton iterative solvers. Building on previous work that used sparse Hessian approximations, the authors introduce low-rank approximation added to the sparse format (hence sparse "plus" low rank in the title). The authors also provide convergence analysis, showing linear convergence.

**Claims And Evidence:**

Authors claim rigorous proof for linear convergence, and demonstrate super-linear convergence in examples.

**Essential References Not Discussed:**

I do not have any suggestions.

**Experimental Designs Or Analyses:**

The experiments are standard benchmarks (MNIST, Fashion-MNIST, ImageNet, etc), I find them suitable for analyzing convergence rates.

**Methods And Evaluation Criteria:**

Authors demonstrate the performance of the new sparse plus low rank Hessian approximated quasi-Newton iterations by applying to benchmark problems. These make sense, although there is some room for more challenging problems.

**Other Comments Or Suggestions:**

minor notes:
- page 1: perhaps mention what a cost matrix is (or its form)?
- The notation $H^{k+1}_\Omega$ (e.g. page 5) is a bit confusing, since powers $(H_\Omega)^{k}$ also appear in the text. Perhaps there is a better version that keeps the iteration number as subscripts?

**Other Strengths And Weaknesses:**

Strengths

The proofs rely on clever combinations of elementary linear algebraic results. Presentation is clear, the proofs are well-written.

Weaknesses

The overall motivations for seeking sparse plus low-rank perhaps can be explained for the benefit of the audience. Is there a more intuitive argument why such an approximation can converge faster?

The authors do not present a potential limitation to the method.

**Questions For Authors:**

What is the overall motivation for using sparse approximations? I understand the low-rank Hessian updates are well-known to be effective in general, but sparse approximations seem special. Adding a sparse matrix to a low-rank one seems to resemble robust PCA; is there any insightful relation between robust PCA and this choice of Hessian approximation?

Should the addition of the low-rank part affect the choice of random selection criteria in sparse directions? i.e. Is there a reason to stick with Algorithm 2 even though now low-rank term was added in the Hessian?

**Relation To Broader Scientific Literature:**

The existing methods have used low-rank approximations to the quasi-Newton iterations (e.g. BFGS) or sparse approximations, but not both. This work uses the sum of both types of approximations in a systematic way to accelerate convergence.

**Theoretical Claims:**

I only did a light check of the proofs, the claims seem to be reasonable and I did not spot any errors.

---

> ### Author Rebuttal · Authors · 2025-04-01
>
> Thank you for your detailed review and thoughtful comments. We give our point-by-point responses below.
>
> ---
>
> ### Weakness 1
>
> Thanks for raising this question on motivation. Our approach is motivated by two key insights:
>
> 1. **Problem-specific Sparsity**: The entropic-regularized optimal transport (OT) problem naturally exhibits (approximatly) sparse Hessian structure, particularly when $\eta$ is small. This sparsity is unique to OT and related problems.
> 2. **General Low-Rank Utility**: Low-rank approximations are successfully applied in quasi-Newton methods (e.g., BFGS, L-BFGS) because they efficiently capture dominant curvature information while maintaining computational tractability.
>
> Our intuition is that the sparse-plus-low-rank (SPLR) structure may better preserve the problem's intrinsic geometry compared to pure sparse or pure low-rank approaches, leading to a faster convergence speed, as is shown in our empirical results. For example, if the Hessian "residuals" (i.e., the elements removed during sparsification) have some low-rank structure, then SPLR naturally captures this information.
>
> ### Weakness 2
>
> We thank you for highlighting this important consideration. Our primary limitation is that although superlinear-like convergence performance is observed, we currently lack formal theoretical justification for this observation. We will clarify such limitation in the revised manuscript.
>
> ### Note 1
>
> We thank you for this helpful suggestion to improve clarity. In the introduction, we will add a brief explanation of the cost matrix in OT problems:
>
> > The cost matrix $C \in \mathbb{R}^{n \times m}$ encodes the pairwise transportation costs between source and target distributions, where $C_{ij}$ represents the cost of moving one unit of mass from source location $i$ to target location $j$. For example, when comparing two discrete probability distributions over spatial positions, $C_{ij}$ might be the Euclidean distance $\| x_i - y_j \|^2$ between points $x_i$ and $y_j$ in the source and target domains, respectively.
>
> We agree that further clarification will help readers, particularly those less familiar with OT, and will incorporate it into the revised manuscript.
>
> ### Note 2
>
> We thank you for this observation, and appreciate the opportunity to clarify our notation system:
>
> Indeed, we already use iteration number as subscripts:
>
> - $H_k$: the true Hessian matrix at iteration $k$;
> - $B_k$: the approximated Hessian matrix with SPLR at iteration $k$.
>
> The point is that we use $H_{\Omega}^{k+1}$ to distinguish between the true Hessian and the sparsified ones. We acknowledge that better notations might be possible, but this is the clearest expression we have developed so far. When we need to express the power of $H$ (which only appears in **Assumption 3.2**), we have added a pair of parentheses around $H$. We will add a remark in the revised paper to avoid potential ambiguity.
>
> ### Question 1
>
> We thank you for this excellent question. Here are our comments:
>
> 1. **Motivation**: See our response to **Weakness 1** above.
> 2. **Connection to robust PCA**: The reviewer astutely observes the conceptual parallel to robust PCA. Indeed, both approaches rely on a _sparse + low-rank_ decomposition to provide a structured approximation of a given matrix. However, while robust PCA—formulated as a _Principal Component Pursuit (PCP)_ problem [1]—aims to _recover_ the low-rank and sparse components through optimization, SPLR adopts a more _constructive_ approach, providing these components via closed-form solutions. Given that robust PCA has been shown to effectively capture the essential structure of a matrix, we argue that SPLR’s approximation should likewise preserve the key information in the Hessian, ensuring a well-justified representation. Importantly, this intuition is further supported by our experimental results, where SPLR shows a superlinear-like convergence speed.
>
> Your comment has helped us recognize that this connection deserves more explicit treatment, and we appreciate the opportunity to elaborate on these important points.
>
> ### Question 2
>
> We thank you for raising this important point. As noted, the low-rank term (Eq. 7) is computed from the sparsified Hessian $H_{\Omega}^{k+1}$​, meaning the random selection in Algorithm 2 directly influences the low-rank approximation.
>
> The rationale behind combining Algorithm 2 with the low-rank term is to strike a balance between computational efficiency and information retention. While Algorithm 2 deliberately sparsifies the Hessian to reduce computational cost—inevitably losing some structural information—the low-rank term serves as a compensatory mechanism. This hybrid approach ensures that we maintain a meaningful approximation to the Hessian while remaining computationally tractable.
>
> [1] Candès, Emmanuel J., et al. "Robust principal component analysis?." Journal of the ACM (JACM) 58.3 (2011): 1-37.

---

### Official Review · Reviewer_2UKW · 2025-03-17

**Overall Recommendation:** 3

**Summary:**

The authors propose a quasi-Newton algorithm for solving entropic optimal transport (EOT) problems. The classical Sinkhorn algorithm enjoys linear convergence with a rate independent of the problem dimension but depends exponentially on the supremum norm of the cost function. There have been some recent works that aim to develop quasi-Newton methods for computing EOTs with super-linear convergence based on the idea of sparsely approximating the Hessian of the Kantorovich dual of EOT. The authors propose a flexible, theoretically justified sparsification scheme, and also combine the Hessian sparsification with low-rank Hessian approximation, similarly done as in L-BFGS. The authors deduce an asymptotic convergence guarantee and linear convergence rate for the proposed algorithm. The proposed algorithm seems to work very well in many synthetic and real-data experiments.

**Claims And Evidence:**

Yes.

**Essential References Not Discussed:**

Guillaume Carlier, "On the Linear Convergence of the Multimarginal Sinkhorn Algorithm" SIAM Journal on OptimizationVol. 32, Iss. 2 (2022)10.1137/21M1410634

**Experimental Designs Or Analyses:**

There are through synthetic and real-data experiments. Also, the additional experiment on the necessity of low-rank approximation provided in the appendix was very helpful.

**Methods And Evaluation Criteria:**

Yes.

**Other Comments Or Suggestions:**

See above

**Other Strengths And Weaknesses:**

**strengths**

1. The paper is relatively well-written and it was easy to follow.
2. The proposed algorithm seems to be performing very well in various tasks, outperforming many existing methods.
3. There are some theoretical guarantees on the Hessian sparsification and convergence guarantee.


**weakness**

1. Theoretical guarantees, especially for the ones for convergence, are weak. Please see the comments in the theory section.
2. L043 "Sinkhorn generality exhibits sub-linear convergence": I don't think this is quire right. Do the authors have reference to back up this claim? Experimentally, in all plots Sinkhorn shows linear convergence. Theoretically, Sinkhorn enjoys linear convergence with a rate independent of the problem dimension but depends exponentially on the supremum norm of the cost function [Car 22]. Also, the authors write in the same paragraph that optimizing on the Kantorovich dual as a different approach than Sinkhorn, but Sinkhorn is just an alternating maximization on the dual and it looks like matrix scaling in the primal space.
3. L125 and hereafter: Please just use $\nabla f(x)$ instead of $g(x)$.
4. L129: $\tilde{T}$ is not defined.
5. L163: The authors state that the existing Hessian sparsification by thresholding by values does not give a control on the density after thresholding. But one could easily threshold the top x % to get a desired density after thresholding, and this is essentially what the authors do.
6. Thm. 4.1: the set $\Omega^{*}$ is ill-defined.
7. The first paragraph in Sec. 4.3 is a repetition from the previous page.
8. Comparing the two plots in Fig. 1 in the second and the third column, it appears that BCD (Sinkhorn) is faster than SPLR in the iteration number. Since it is the cheapest per-iteration, it should be much faster than SPLR than in wall-clock time. The plots in the second row are misleading since SSNS is too slow and jams all other curves.
9. L954: One needs to argue the sub-level set D is compact in order to obtain uniform bounds L,U on the eigenvalues of the Hessian over D. This is missing.

**Questions For Authors:**

See above

**Relation To Broader Scientific Literature:**

The method for computing EOT provided in this work could potentially be very useful for other researchers and practitioners.

**Theoretical Claims:**

**Thm. 3.3 and Cor. 3.4 on Hessian sparsification**

1. The authors state that these results are significant because they guarantee positive definiteness of the Hessian after sparsification. But since they only sparsify off-diagonal entries and all Hessian entries are positive, diagonal dominance (or Gershgorin's circle theorem) trivially implies that the Hessian remains positive definite after any specification pattern. It does not, however, directly imply that the condition number improves. I think this latter point should be the main emphasis of these results.

2. In the second and the third remark, the authors point out that their incremental sparsification decreases the Hessian's condition number, so it leads to numerical stability. While this is certainly true, there is no quantitative bound on the improvement of the condition number. If it is hard to justify theoretically, I recommend validating this claim by adding some experiments on plotting the condition number vs. iteration.

3. It is not clear why Assumption 3.2 is "very weak" at this point.


**Thm. 5.1, Cor. 5.2, Thm. 5.3 on the convergence guarantees**

1. These results are derived from standard analysis of quasi-Newton methods, as the authors indicate in the appendix. Namely, the authors take the quotient space of the dual variables by setting $\beta_{n}=0$ to get rid of the shift-invariance of the dual objective ($f(x)$ in their notation). While the dual objective is only convex and not strongly convex originally, after constraining on this subspace it becomes strictly convex. Further restricting this on the level set, one can obtain uniform bounds on the eigenvalues of the Hessian along the trajectory of the algorithm, provided that the objective values decrease monotonically.

However, this standard analysis does not give linear convergence rate that is independent of the problem dimension. For instance, the rate $r$ in Thm. 5.3 may depend on $m$ and $n$, so the linear convergence stated there could effectively be sublinear for large-scale problems. The issue is that the condition number of the dual objective can be as large as order n. For instance, if at the current iterate the dual variables $\alpha_k$ and $\beta_k$ are the same (assuming $m=n$), then the $2n \times 2n$ full Hessian has zero eigenvalue since the diagonal entries are $Cn$, which is the absolute row sums of the off-diagonal entries. Imposing $\beta_{n}$ gets rid of one row from the (1x2) block matrix and the corresponding column of the (2 x 1) block, and the resulting off-diagonal sum is $C(n-1)$. The maximum eigenvalue is still of order n, but the minimum eigenvalue is of order 1.

In the state-of-the-art analysis of Sinkhorn (e.g., Carlier 2022 SIOPT), one addresses such issue by directly showing that the Sinkhorn iterates are uniformly bounded (by using closed-form expression of Sinkhorn iterates) and strong convexity of the entry-wise dual objective (in this case the exponential function) on a compact interval. Can the authors adapt a similar strategy and get a dimension-independent linear convergence rate of their algorithm?

---

> ### Author Rebuttal · Authors · 2025-04-01
>
> Thank you for your detailed review and thoughtful comments. We give our point-by-point responses below.
>
> ---
>
> ### Thm.3.3 and Cor.3.4
>
> 1. We respectfully clarify that positive definiteness cannot be directly guaranteed via the Gershgorin circle theorem in this case. Specifically, for the sparsified Hessian $H_{\Omega^*}$, the Gershgorin disc centered at $h_{n+1,n+1}$ has radius $\sum_{j \neq n+1} |h_{n+1,j}|$, admitting $\lambda = 0$ as a possibility. Thus, the theorem alone does not preclude singularity. Moreover, you are correct that explicitly highlighting the improvement in the condition number would strengthen the impact of our results and we will emphasize this point more clearly.
>
> 2. Thanks for this insightful suggestion. Indeed we have conducted numerical experiments on the condition number (this was actually done before we develop the theory), and we will add such experiments in the revised paper.
>
> 3. There are two key reasons:
>
>    1. The assumption is analogous to the concept of *irreducibility* in Markov chains. Irreducibility is considered a mild condition because it only requires connectivity, not quantitative bounds on probabilities and it is *necessary but not sufficient* for stronger properties (e.g., ergodicity).
>
>    2. Assum. 3.2 can be satisfied with *extremely sparse* matrices—_e.g._, retaining just one complete row and column (yielding a density lower bound of only $O(1/(nm))$).
>
>    We will add a brief remark in the revision to make this intuition clearer.
>
> ### Thm. 5.1, Cor. 5.2, Thm. 5.3
>
> Indeed the current theoretical analysis still follows the classical framework of quasi-Newton methods, but we emphasize that one key ingredient of proving the linear convergence is to bound the condition number of the approximate Hessian matrix, and in our case it is highly related to Thm.3.3 and Cor.3.4, which are unique to OT problems. Your invaluable comments have encouraged us to actively explore novel technical tools to further optimize the linear rate constant.
>
> We will revise the paper to acknowledge the significance of the theory in Carlier (2022).
>
> ### Weakness 1
>
> See our responses above.
>
> ### Weakness 2
>
> You are absolutely correct that the Sinkhorn algorithm has a linear convergence. Our phrasing was indeed misleading, and we will correct it in the revision. Our intended point was that, while the convergence is *theoretically* linear, the rate can *practically* resemble sub-linear behavior when the regularization parameter $\eta$ is small ($1 - e^{-24\\|C\\|_{\infty}/\eta}$ approaches 1).
>
> Also, Sinkhorn is indeed an alternating maximization procedure on the dual. We will revise for precision.
>
> ### Weakness 3
>
> We will update the manuscript accordingly.
>
> ### Weakness 4
>
> $\tilde{T}$ represents the first $(m-1)$ columns of matrix $T$, as defined in **Notation** (Line 68).
>
> ### Weakness 5
>
> While thresholding the top x% of elements would indeed allow direct density control, we emphasize that existing Hessian sparsification methods do not employ this approach, nor do they provide theoretical guarantees for arbitrary density selection. Specifically:
>
> 1. SNS: Uses an elementwise threshold $\rho$. Though we can set the Hessian to a desired density, there are no theoretical gurantees on invertibility or convergence rate.
>
> 2. SSNS: Sets a Hessian error threshold $\delta_k = \nu_0 \\|g_k\\|^{\gamma}$ at each iteration, which depends on the gradient norm for theoretical guarantees. This scheme cannot be set to an arbitrary density a priori, as $\delta_k$ varies with optimization progress.
>
> In contrast, our proposed scheme explicitly enables any desired density level.
>
> ### Weakness 6
>
> The sparsification pattern $\Omega^*$ consists of all coordinates in the first row and the first column.
> For example, if $n=4$ and $m=4$, then $\Omega^*=\\{(1, 1), (1, 2), (1, 3), (2, 1), (3, 1), (4, 1)\\}$.
>
> Could you please clarify in what specific way $\Omega^*$ appears ill-defined? We would be happy to provide additional explanations.
>
> ### Weakness 7
>
> We sincerely apologize for this unintentional redundancy and we will carefully revise this section to eliminate repetition.
>
> ### Weakness 8
>
> The discrepancy in our current plots likely stems from that each Sinkhorn "iteration" in our implementation includes both the $\alpha$ and $\beta$ optimization steps (two full matrix scaling operations).
>
> ### Weakness 9
>
> We thank you for this important technical observation. We will make the proof more rigorous in the revised manuscript. Below are the main ideas: we can first show that the optimal function value is finite, i.e., $f^*>-\infty$. Then the set $D^*=\\{x:f(x)\le f^*\\}=\\{x^*\\}$ is non-empty and bounded. Corollary 8.7.1 of [1] shows that $D_c=\\{x:f(x)\le c\\}$ is bounded for every $c$, which implies that $D = \\{x: f(x) \leq f(x_0)\\}$ is bounded.
>
> [1] R Tyrrell Rockafellar. Convex Analysis. Princeton University Press, 1970.

---

### Decision · Program_Chairs · 2025-05-01

**Decision:**

Accept (poster)

**Comment:**

The paper introduces a novel quasi-Newton method that exploits sparse plus low-rank Hessian approximations for entropic OT, demonstrating superior empirical performance over existing methods (Sinkhorn, L-BFGS, etc.) in both synthetic and real-data experiments. While some theoretical gaps remain (e.g. dimension-dependent rates, superlinear convergence in practice), the method’s practical efficacy and solid analysis leads me to recommend that the paper should be accepted.